# RING finger protein 13 protects against nonalcoholic steatohepatitis by targeting STING-relayed signaling pathways

Zhibin Lin[1,3], Peijun Yang[1,3], Yufeng Hu[2,3], Hao Xu[1], Juanli Duan[1], Fei He[1], Kefeng Dou[1] ✉ & Lin Wang [1] ✉

Nonalcoholic fatty liver disease (NAFLD) is the most common liver disorder worldwide. Recent studies show that innate immunity-related signaling pathways fuel NAFLD progression. This study aims to identify potent regulators of innate immunity during NAFLD progression. To this end, a phenotype-based high-content screening is performed, and RING finger protein 13 (RNF13) is identified as an effective inhibitor of lipid accumulation in vitro. In vivo gain- and loss-of-function assays are conducted to investigate the role of RNF13 in NAFLD. Transcriptome sequencing and immunoprecipitation-mass spectrometry are performed to explore the underlying mechanisms. We reveal that RNF13 protein is upregulated in the liver of individuals with NASH. *Rnf13* knockout in hepatocytes exacerbate insulin resistance, steatosis, inflammation, cell injury and fibrosis in the liver of diet-induced mice, which can be alleviated by *Rnf13* overexpression. Mechanically, RNF13 facilitates the proteasomal degradation of stimulator of interferon genes protein (STING) in a ubiquitination-dependent way. This study provides a promising innate immunity-related target for NAFLD treatment.

Nonalcoholic fatty liver disease (NAFLD), the most common liver disorder worldwide, encompasses a spectrum of diseases spanning from simple steatosis or nonalcoholic fatty liver (NAFL) to nonalcoholic steatohepatitis (NASH), NASH-associated fibrosis, cirrhosis, and hepatocellular carcinoma[1–3]. Left untreated, NAFL may progress into NASH, resulting in a high risk of liver failure. It is estimated that NASH is likely to become the leading indication for liver transplantation within a decade[1]. Despite the tremendous socioeconomic burden NAFLD imposes, effective therapeutic interventions for NAFLD/NASH have not been approved[4]. Hence, a full understanding of the disease pathogenesis is urgently needed.

Emerging evidence proves that the activation of signaling pathways associated with innate immunity is one of the driving forces in NAFLD[5–7]. The initiation of these pathways begins with the recognition of pathogen- or damage-associated molecular patterns (PAMPs/ DAMPs) by membrane-bound and intracellular pattern-recognition receptors (PRRs), such as toll-like receptors (TLRs) and nucleotide-binding oligomerization domain (NOD)-like receptors (NLRs)[8]. Then, several key adaptor proteins such as myeloid differentiation factor-88 (MyD88), mitochondrial antiviral signaling protein (MAVS), and stimulator of interferon genes protein (STING) could relay the signaling, triggering the activation of transcription factors and production of proinflammatory cytokines[6]. Components of the innate immune system have been proven to widely engage in the pathophysiology of NAFLD[5–7]. In the latest study, multispecies transcriptome profiling of NASH liver reveals a dominant role of innate immune signaling pathways in NASH pathogenesis[9]. Moreover, similar to conventional inflammatory cells, hepatocytes and liver sinusoidal endothelial cells (LSECs) can exert immunological functions upon metabolic disturbance[10]. Therefore, restricting the abnormal activation of innate

[1]Department of Hepatobiliary Surgery, Xi-Jing Hospital, Fourth Military Medical University, Xi'an 710032, China. [2]Gannan Innovation and Transformation Medical Research Institute, First Affiliated Hospital of Gannan Medical University, Gannan Medical University, Ganzhou 341000, China. [3]These authors contributed equally: Zhibin Lin, Peijun Yang, Yufeng Hu. ✉e-mail: doukef@fmmu.edu.cn; fierywang@163.com

immunity-related signaling pathways confers great importance to NAFLD treatment.

Ubiquitination has been well documented to control the signaling cascades of innate immunity and fine-tune the production of proinflammatory cytokines[11–13]. Hence, it is tempting to speculate that the main conductors of ubiquitination (E3 ubiquitin ligases and deubiquitinating enzymes, DUBs) that have been reported to regulate innate immune signaling, can also control NAFLD progression. To dig out such regulators, we performed a phenotype-based high-content screening and analysis. Finally, we identified an E3 ligase RING finger protein 13 (RNF13) as a potent regulator of lipogenesis and inflammation response in hepatocytes.

RNF13 and eight other proteins belong to the Goliath family, based on their high similarity in the PA-TM-RING structure − a protease-associated domain (PA), a transmembrane domain (TM), and a RING finger domain[14]. Studies have proved that most of the family members are involved in the regulation of the immune system[15–18]. In the present study, we prove that RNF13 can control lipid deposition and inflammation response in NAFLD by regulating the level of two components of the innate immunity system, which are tripartite motif-containing 29 (TRIM29) and STING.

## Results

### RNF13 protects hepatocytes from PAOA-induced lipid accumulation and inflammation

In order to dig out potent innate immunity-associated E3 ligases or DUBs in NAFLD progression, we constructed 50 plasmids, according to current literatures[11,12,15]. Then we transfected them into HepG2 and Huh7 cell lines, followed by palmitic acid and oleic acid (PAOA) treatment. After Nile Red staining, we analyzed the impacts of these plasmids on PAOA-induced lipid deposition in hepatocytes via a high-content imaging system. Results showed that 11 plasmids significantly affected lipid accumulation in HepG2 cell line (Supplementary Fig. 1a), while 23 plasmids significantly mitigated lipid accumulation in Huh7 cell line (Supplementary Fig. 1b). After integrating the results, we noted that an E3 ligase RNF13, which have not been investigated in NAFLD, played an obvious protective role in lipid deposition (Supplementary Fig. 1c). Hence, we selected RNF13 as the candidate molecule. Next, we evaluated the role of RNF13 in mouse primary hepatocytes (MPHs) by the use of *Rnf13*-overexpressing adenovirus (Ad*Rnf13*) and knockdown adenovirus (Adsh*Rnf13*). Nile Red staining indicated that PAOA-induced lipid accumulation in hepatocytes was intensified in Adsh*Rnf13*-infected hepatocytes but not AdshRNA-infected ones (Supplementary Fig. 2c, d). Consistently, RNF13 knockdown significantly increased the triglyceride (TG) concentration in PAOA-treated primary hepatocytes (Supplementary Fig. 2e). The expression of lipogenic and proinflammatory genes in PAOA-challenged hepatocytes was also upregulated after RNF13 knockdown (Supplementary Fig. 2f, g). While RNF13 overexpression attenuated the lipid deposition and inflammatory response in PAOA-treated hepatocytes (Supplementary Fig. 2h–n). Overall, RNF13 exerts a protective role in hepatocytes upon PAOA treatment.

### Hepatic RNF13 expression is induced in NASH pathogenesis

Next, we detected RNF13 expression in NASH livers. Immunohistochemistry staining showed an increase of RNF13 expression in hepatocytes from the liver sections exhibiting pathological features of NASH (Fig. 1a), which was confirmed by western blot analysis (Fig. 1b). And RNF13 protein level positively correlated with NAFLD activity score (NAS) (Fig. 1c). However, RNF13 mRNA level remained unchanged in the NASH livers (Fig. 1d). Thereafter, we investigated the RNF13 expression in different NAFLD-associated models. We observed an increased tendency of RNF13 protein in the livers of mice fed with a high-fat diet (HFD) and a high-fat and high-cholesterol (HFHC) diet, with the HFHC group presenting the highest RNF13 protein level

(Fig. 1e). Whereas the mRNA level was not upregulated (Fig. 1f). We further specified which cell type contributing to the upregulation of RNF13 in NASH. We observed that RNF13 protein but not mRNA was significantly increased in MPHs and the HepG2 cell line treated with PAOA (Fig. 1g-j), with no obvious alteration in nonparenchymal cells after corresponding pathogenic stimulation (Fig. 1k, l). Collectively, RNF13 protein is induced in hepatocytes during NASH progression.

### RNF13 protein becomes more stable upon PAOA stimulation

We then investigated the mechanisms accounting for the upregulation of RNF13 during NAFLD progression. Considering that RNF13 mRNA did not increase in NASH, and previous study indicates that RNF13 protein undergoes extensive post-translational proteolysis in both lysosome and proteasome, which makes it rather unstable[14], we first compared the stability of RNF13 in the setting of BSA or PAOA treatment by cycloheximide (CHX) chase assays. As shown in Fig. 2a, RNF13 protein underwent less severe degradation and had a longer half-life in PAOA-treated hepatocytes than in BSA-treated hepatocytes. And RNF13 degradation was primarily governed by lysosome, since the lysosome inhibitor (chloroquine, CQ) but not the 26 S proteasome inhibitor (MG132) could rescue RNF13 protein level in the setting of PAOA (Fig. 2b, c). Therefore, it is plausible that the lysosomal degradation of RNF13 is inhibited upon PAOA treatment. In testifying it, we confirmed that RNF13 co-localizes well with lysosomal-associated membrane protein 1 (LAMP1) in BSA-treated hepatocytes, whereas this colocalization was reduced by PAOA treatment (Fig. 2d); CQ treatment increased RNF13 protein abundance in BSA-treated hepatocytes to the level in PAOA-treated ones (Fig. 2e). Since previous studies have indicated the intense auto-ubiquitination of RNF13[14,19], and ubiquitination has been documented to directs internalized proteins toward lysosome[20], we wondered whether the lysosomal degradation of RNF13 was governed by ubiquitination. Therefore, we compared RNF13 ubiquitination with BSA and PAOA treatment. Result showed that obvious ubiquitination happened on RNF13 in BSA treatment, which markedly decreased upon PAOA stimulation (Fig. 2f). Screening for potential lysine ubiquitination types revealed that PAOA primarily inhibited the attachment of K63O (ubiquitin with the intact Lys63 residue alone) to RNF13 (Fig. 2g). Similarly, other studies have revealed the positive impact of K63-linked ubiquitination on lysosomal degradation[20–22]. To further confirm it, we constructed the RNF13 mutant (C258A/H260A) plasmid, in which cysteine (C258) and histidine (H260) in the ring-finger domain were mutated to alanine, resulting in the loss of its E3 ligase activity (Fig. 2h). RNF13 mutant has a longer half-life than RNF13 wild type in BSA-treated MPHs, and the PAOA stimulation did not affect the half-life of RNF13 mutant (Fig. 2i). Consistently, RNF13 mutant did not co-localize with LAMP1 in BSA or PAOA setting (Fig. 2j). In summary, we prove that RNF13 undergoes extensive K63-linked auto-ubiquitination and subsequent lysosomal degradation in normal circumstances, whereas PAOA stimulation can alleviate the auto-ubiquitination and degradation, making RNF13 more stable (Fig. 2k).

### *Rnf13*[HKO] mice present more severe insulin resistance, hepatic steatosis and liver injury in the HFD model

To detect the functions of RNF13 in vivo, we generated RNF13 hepatocyte-specific knockout (*Rnf13*[HKO]) mice as well as their counterparts *Rnf13*[Flox/Flox] by application of CRISPR-Cas9 (Supplementary Fig. 3a–c). *Rnf13*[HKO] and *Rnf13*[Flox/Flox] mice were fed with either a NCD or a HFD for 24 weeks (Fig. 3a). Despite no significant difference in body weight was observed in NCD- or HFD-fed *Rnf13*[HKO] and *Rnf13*[Flox/Flox] mice (Fig. 3b), *Rnf13*[HKO] mice presented abnormal glucose metabolism after HFD feeding. As shown in Fig. 3c–d, no grossly visible hyperglycemia was shown in *Rnf13*[HKO] mice before HFD feeding, whereas *Rnf13*[HKO] mice showed higher glucose levels compared with *Rnf13*[Flox/Flox] littermates after HFD feeding. Moreover, *Rnf13*[HKO] mice manifested worse

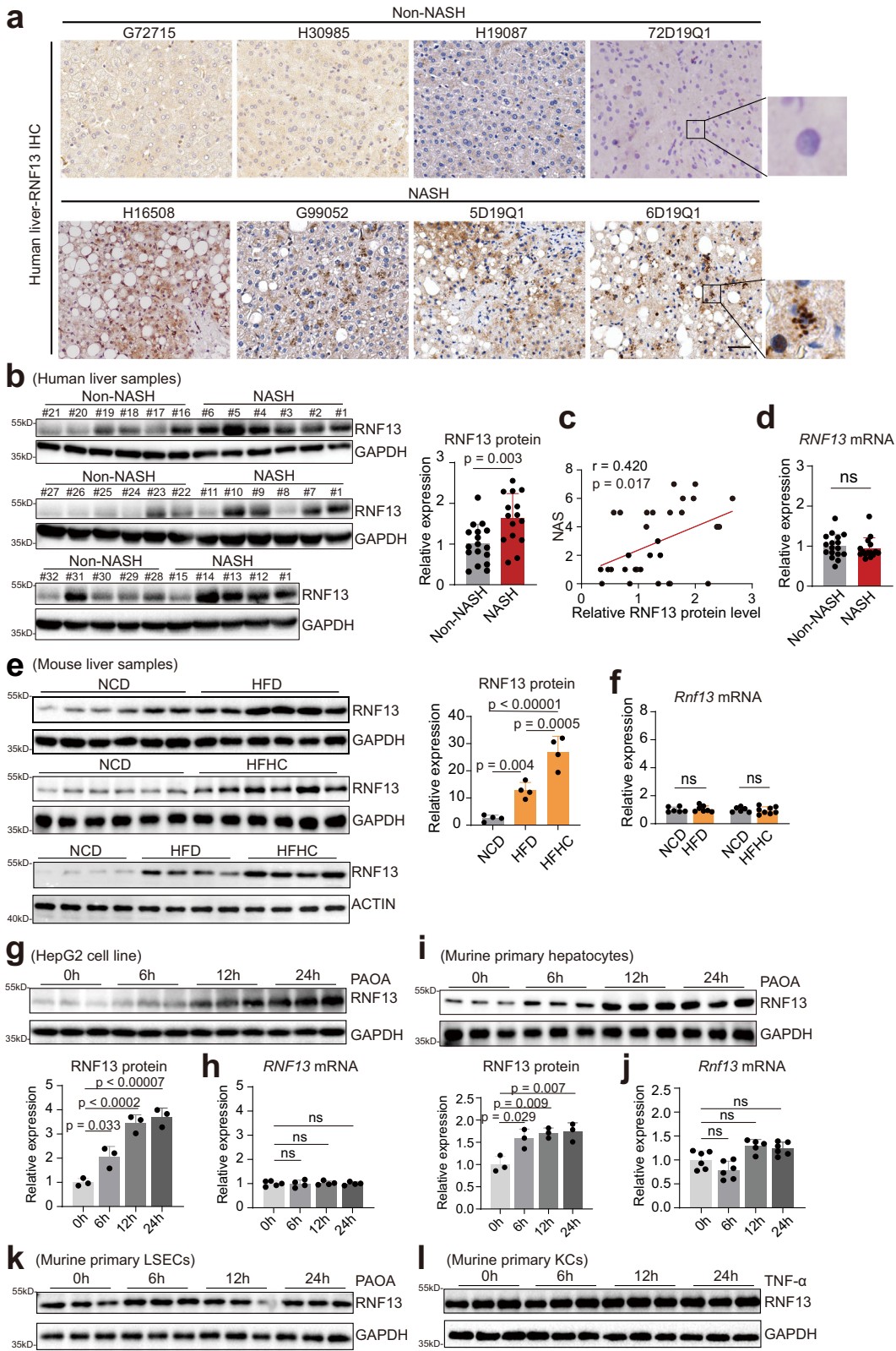

**Fig. 1 | Induced RNF13 in NAFLD progression. a** Immunohistochemistry staining of RNF13 in liver biopsies from non-NASH and NASH individuals ($n = 4$). Scale bars, 50 μm. RNF13 protein (**b**) and mRNA (**d**) expression in human liver samples from non-NASH ($n = 17$) and NASH ($n = 15$) individuals. Human samples were derived from the same experiment and blots were processed in parallel. **c** Correlation analysis for RNF13 protein levels and NAFLD activity score (NAS) in human liver samples ($n = 32$). RNF13 protein (**e**) and mRNA level (**f**) in liver samples from NCD, HFD, and HFHC-fed mice (For **e**, $n = 4$, 6; for **f**, $n = 6$ in the NCD group, $n = 7$ in the HFD group, $n = 8$ in HFHC group). RNF13 protein (**g**) and mRNA expression (**h**) in HepG2 cells

treated with PAOA for indicated hour. RNF13 protein (**i**) and mRNA expression (**j**) in murine primary hepatocytes (MPHs) treated with PAOA for indicated hour (For **g** and **i**, $n = 3$; for **h**, $n = 5$ in 0 h group, $n = 4$ in other groups; for **j**, $n = 5$ in 12 h group, $n = 6$ in other groups). RNF13 protein expression in murine primary LSECs treated with PAOA (**k**) and Kupffer cells treated with TNF-α (**l**) for indicated hour ($n = 3$). Data were expressed as mean ± SD. Two-tailed Student's *t*-test for **b**, **d**, and **f**, one-way ANOVA with Bonferroni post hoc analysis for **e** and **g–j**, Spearman correlation analysis for **c**. Source data are provided as a Source Data file.

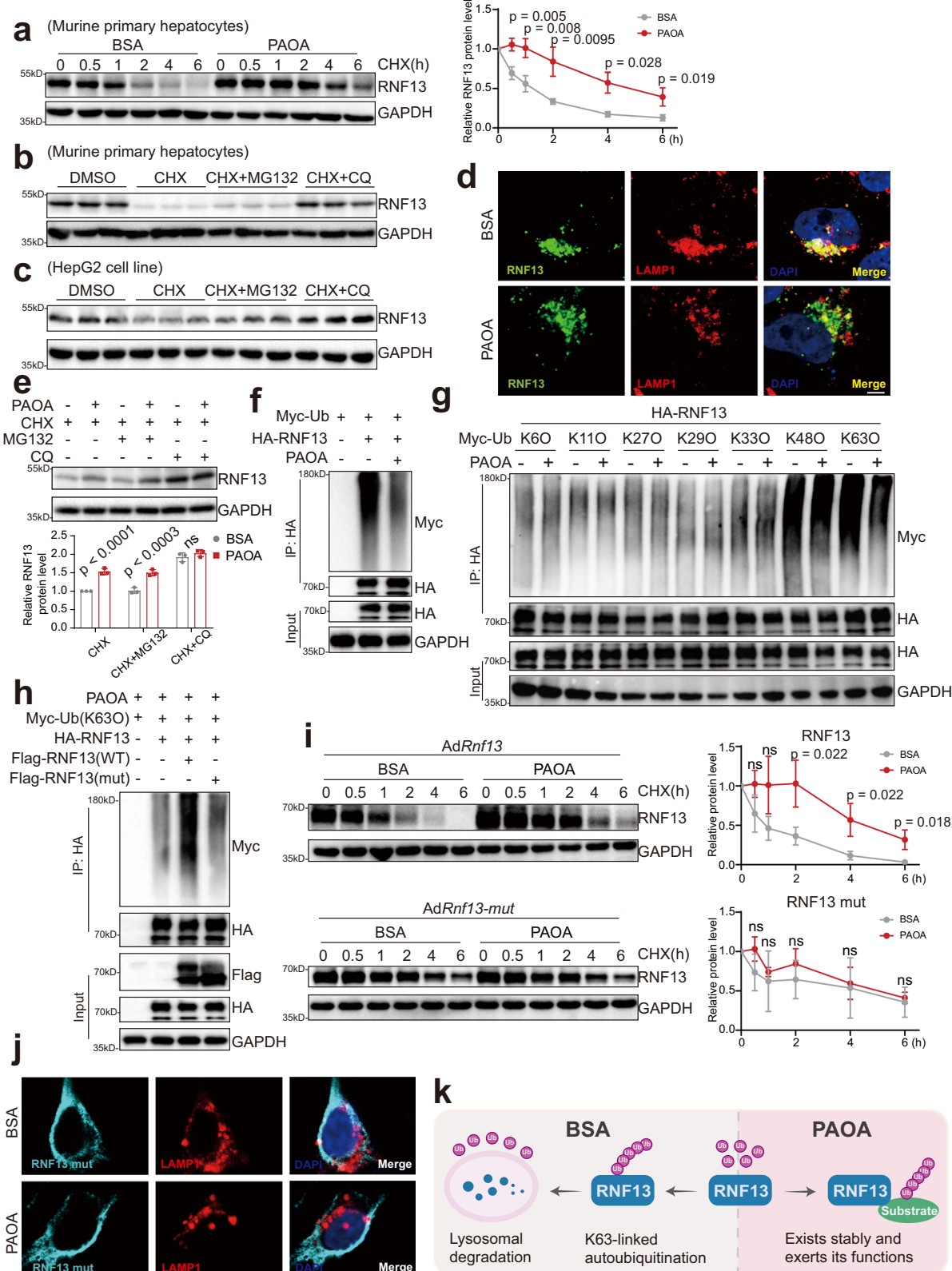

glucose tolerance, as indicated by the glucose tolerance test (GTT) (Fig. 3e, f), and impaired insulin resistance, as indicated by the insulin tolerance test (ITT) (Fig. 3g, h). *Rnf13* knockout also caused a disturbance of lipid metabolism. On one hand, we observed an increase in serum total cholesterol (TC) and triglyceride (TG) in *Rnf13*^HKO mice after HFD feeding (Fig. 3i, j). On the other hand, HFD feeding induced more obvious steatosis in the liver of *Rnf13*^HKO mice compared with

*Rnf13*^Flox/Flox littermates, as shown by higher liver weight or liver-to-body weight ratio (Fig. 3k, l), and more lipid droplets in the liver (Fig. 3m–o). Analysis of lipometabolic gene expression also validated the impact of RNF13 on hepatic steatosis (Fig. 3p). Furthermore, *Rnf13*^HKO mice challenged with HFD suffered more serious liver injury than the control mice, since higher alanine aminotransferase (ALT) and aspartate aminotransferase (AST) levels were detected in *Rnf13*^HKO

**Fig. 2 | RNF13 becomes more stable upon PAOA stimulation. a** RNF13 protein expression in MPHs treated with BSA or PAOA plus CHX for indicated hour. RNF13 protein expression in MPHs (**b**) and HepG2 cells (**c**); cells were treated with DMSO, CHX, CHX plus MG132 or CHX plus CQ for 6 h in the presence of PAOA. **d** Immunofluorescent staining of exogenous RNF13 (green) and endogenous LAMP1 (red) in HepG2 cells treated with BSA or PAOA for 12 h. Nuclei were counterstained with DAPI. Scale bars, 5 μm. **e** RNF13 protein expression in MPHs treated with CHX, CHX plus MG132 or CHX plus CQ for 6 h in the setting of BSA or PAOA. **f, g** Ubiquitination of HA-RNF13 in HepG2 cells co-transfected with the indicated plasmids following BSA/PAOA plus CQ treatment. **h** Ubiquitination of HA-RNF13 in HepG2 cells co-transfected with the indicated plasmids following PAOA plus CQ treatment. **i** Wild type or mutant RNF13 expression in MPHs infected with Ad*Rnf13* (wild type, upper panel) or Ad*Rnf13* (mutant, lower panel) following BSA or PAOA plus CHX treatment for indicated hour. **j** Immunofluorescent staining of exogenous RNF13 mutant (cyan) and endogenous LAMP1 (red) in HepG2 cells treated with BSA or PAOA for 12 h. Nuclei were counterstained with DAPI. Scale bars, 5 μm. **k** A schematic diagram showing the state of RNF13 with or without PAOA stimulation. For **a**–**j**, at least three independent experiments have been conducted. Data were expressed as mean ± SD. Two-tailed Student's *t*-test for **a** and **i**, one-way ANOVA with Bonferroni post hoc analysis for **e**. Source data are provided as a Source Data file.

mice (Fig. 3q, r). Overall, RNF13 suppression aggravates HFD-induced insulin resistance, hepatic steatosis, and liver injury.

### *Rnf13*[HKO] mice present more severe hepatic steatosis, inflammation and fibrosis in the HFHC model

We further examined whether RNF13 participates in NASH etiology by challenging *Rnf13*[HKO] and *Rnf13*[Flox/Flox] mice with a HFHC diet (Fig. 4a). Consistent with the phenotypes in the HFD-induced model, *Rnf13* knockout intensified insulin resistance, hepatic steatosis, and liver injury in the HFHC-induced NASH model (Figs. 4b–n and 4s–t). Given that HFHC diet-induced NASH model can drive more severe inflammation response and fibrosis than HFD[2], we measured the inflammation- and fibrosis-related indicators. CD11b staining of liver sections revealed more intensified infiltration of inflammatory cells in the liver of *Rnf13*[HKO] mice than the *Rnf13*[Flox/Flox] mice after HFHC feeding (Fig. 4o). And *Rnf13* knockout statistically activated the expression of proinflammatory gene (Fig. 4p). On the other side, *Rnf13* slicing favored more distinct collagen deposition in the liver, as indicated by Picric Sirius Red (PSR) staining, and more active transcription of profibrotic genes, as indicated by qPCR analysis (Fig. 4q, r). To systematically profile the gene expression signature of *Rnf13* knockout in NASH, we conducted transcriptome sequencing by the use of liver tissues separated from HFHC-fed *Rnf13*[HKO] mice and their counterparts. Hierarchical clustering analysis revealed the separated gene expression profiles between the two genotypes (Fig. 4u). RNF13 knockdown induced the differential expression of 777 genes (DEGs), a lot of which were involved in lipid metabolism, inflammation, fibrosis and apoptosis (Fig. 4v, w). In summary, *Rnf13* knockout exacerbates NASH phenotypes in the HFHC-induced mice.

### *Rnf13*[HepTg] mice present less severe NASH phenotypes in the HFHC model

To further confirm the impacts of RNF13 on NASH progression, we also generated hepatic *Rnf13*-overexpressed transgenic mice (*Rnf13*[HepTg]) and non-transgenic mice (*Rnf13*[NTg]) via the Sleeping Beauty transposon system[23] (Supplementary Fig. 3d). As expected, RNF13 overexpressing significantly ameliorated glucose disturbance after HFHC feeding (Fig. 5c–f), without obvious change in the body weight (Fig. 5b). In the liver, RNF13 overexpressing significantly inhibited the disturbance of lipid metabolism and immune response, and finally mitigated the severe fibrosis and liver injury induced by HFHC feeding (Fig. 5g–u).

### RNF13 inhibits STING-mediated inflammatory signaling pathways in NASH

To investigate the way RNF13 inhibits NASH progression, we performed the Kyoto Encyclopedia of Genes and Genomes (KEGG) analysis and Gene Set Enrichment Analysis (GSEA) on the transcriptome. Results showed that RNF13 suppressing disturbed the signaling transducing of pathways involved in lipogenesis, inflammation, fibrosis and apoptosis, with the cytosolic DNA-sensing pathway, also named cyclic GMP-AMP synthase (cGAS)-stimulator of interferon genes (STING) pathway[24], ranked at the top of the list (Fig. 6a, b). The activated cGAS-STING pathway in liver myeloid

cells has been identified as a novel driver of NASH progression[25,26], whereas its role in hepatocytes remained contradictory. On one hand, previous studies reported that STING is less important in hepatocytes[25,26]; on the other hand, STING in hepatocytes has been proven to induce lipid accumulation as well as inflammation[27–29], and promote liver injury and fibrosis[30–32]. Therefore, we decided to verify the impacts of STING on hepatocytes via a modified cellular model. In this model, mouse primary hepatocytes were infected with Ad*Sting1*, and the culture medium was changed in 6 h. After another 10 h, nonparenchymal cells (mostly Kupffer cells) plated in transwell chambers as well as PAOA were added for a 12-hour stimulation. Finally, hepatocytes were collected for further analyses. We observed that STING overexpression in hepatocytes significantly exacerbated abnormal lipid accumulation induced by PAOA, accompanied by transcriptional alteration of genes associated with inflammation and lipogeneses (Supplementary Fig. 4a–c). Therefore, we considered the cGAS-STING pathway might be the candidate downstream pathway of RNF13 and proceeded to verify. We first performed western blot analyses and results showed that in vitro and in vivo hepatocyte RNF13 suppression induced an increase in STING expression, the activation of two major effector pathways of STING, namely the NF-κB and TBK1-IRF3 signaling pathways[33] (Fig. 6c, e and Supplementary Fig. 4d), and the transcription of interferon-β (IFN-β) (Supplementary Fig. 4e, f, h). Similarly, we observed a decrease in STING protein level, the activity of NF-κB and TBK1-IRF3 signaling (Fig. 6d, f), and IFN-β mRNA level (Supplementary Fig. 4g, i) in RNF13-overexpressed hepatocytes. We then observed that RNF13 downregulated STING protein in a dose-dependent manner, whereas its mRNA was unaffected (Fig. 6g–j). To verify whether RNF13 regulated NASH by reducing the abundance of STING, we conducted rescue experiments. We observed that adenovirus-mediated STING overexpression on one hand restored the activation of the NF-κB and TBK1-IRF3 signaling in Ad*Rnf13*-infected hepatocytes (Fig. 6k); on the other hand, it reverted the lipid deposition and inflammation response that were previously ameliorated by RNF13-overexpressing (Fig. 6l-o). Moreover, the phenotypic changes resulting from RNF13-knockdown can also be reversed by C176, a selective inhibitor of STING[34] (Supplementary Fig. 5a-d). To further confirm the regulatory effects of RNF13 on STING, we applied two kinds of adeno-associated virus serotype 8 (AAV8), namely AAV8-TBG-ZsGreen-*Rnf13* and AAV8-TBG-mCherry-*Sting1*. Both of them carry the thyroxine-binding globulin (TBG) promoter to achieve hepatocyte-specific RNF13 or STING overexpression (Fig. 7a, b). After the 16-week HFHC feeding, we observed that RNF13 overexpression significantly attenuated the abnormal blood glucose (Fig. 7d–f), lipid accumulation in serum as well as liver (Fig. 7g–k, n, o and Supplementary Fig. 5e), inflammation response (Fig. 7p and Supplementary Fig. 5f), fibrosis (Fig. 7q and Supplementary Fig. 5g) and liver injury (Fig. 7l, m). And the therapeutic effects were abolished in the RNF13&STING-overexpressing mice (Fig. 7d–q), indicating that RNF13 ameliorates NASH by regulating STING protein abundance.

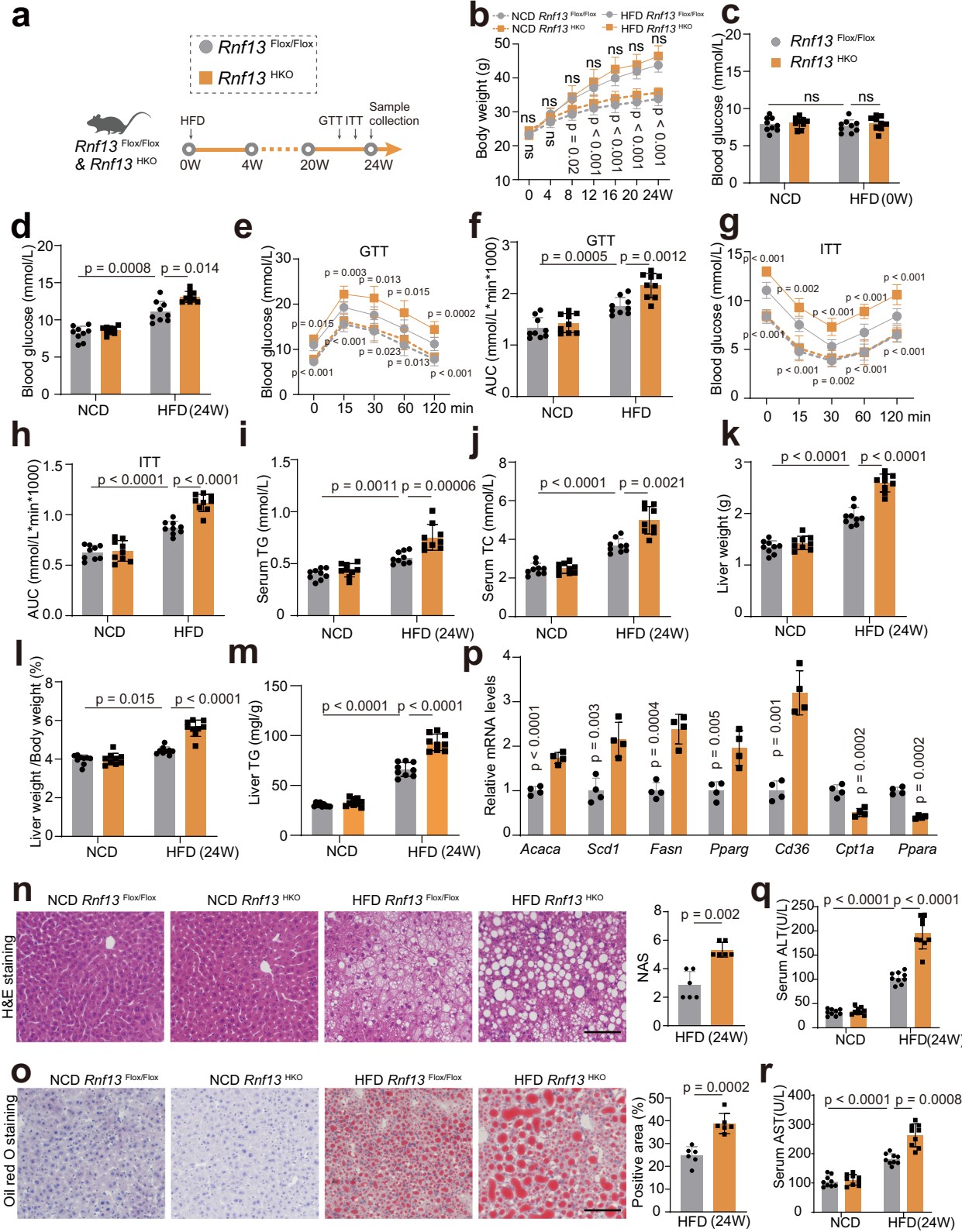

**Fig. 3 | *Rnf13*^HKO mice present more severe insulin resistance, hepatic steatosis and liver injury in the HFD model. a** Schematic depiction of in vivo experiments performed to evaluate the function of RNF13 using hepatocyte-specific *Rnf13* knockout (*Rnf13*^HKO) and the control (*Rnf13*^Flox/Flox) mice fed with a high-fat diet (HFD). Body weights (**b**), blood glucose levels (**c, d**), GTT and ITT assays and the corresponding AUC (**e–h**), serum TG (**i**), serum TC levels (**j**), liver weights (**k**), ratios of liver weight to body weight (**l**) and liver TG levels (**m**) of *Rnf13*^Flox/Flox and *Rnf13*^HKO mice at the indicated time points during NCD or HFD consumption (*n* = 9). H&E (**n**), Oil Red O staining (**o**), and corresponding quantification of liver sections obtained from *Rnf13*^Flox/Flox and

*Rnf13*^HKO mice fed with NCD and HFD for 24 weeks. Scale bars, 100 μm (*n* = 6). **p** Lipometabolic mRNA expression in the liver of *Rnf13*^Flox/Flox and *Rnf13*^HKO mice after HFD feeding (*n* = 4). Serum ALT (**q**) and AST (**r**) levels of *Rnf13*^Flox/Flox and *Rnf13*^HKO mice after NCD or HFD consumption (*n* = 9). Data were expressed as mean ± SD. Two-tailed Student's *t*-test for **n–p**, one-way ANOVA with Bonferroni post hoc analysis for **c, d, f, h–m, q** and **r**, two-way repeated-measures ANOVA followed by Bonferroni post hoc analyses for **b, e** and **g** (Upper *p*-value for comparison between HFD *Rnf13*^Flox/Flox group and HFD *Rnf13*^HKO group; Lower *p*-value for comparison between NCD *Rnf13*^Flox/Flox group and HFD *Rnf13*^Flox/Flox group). Source data are provided as a Source Data file.

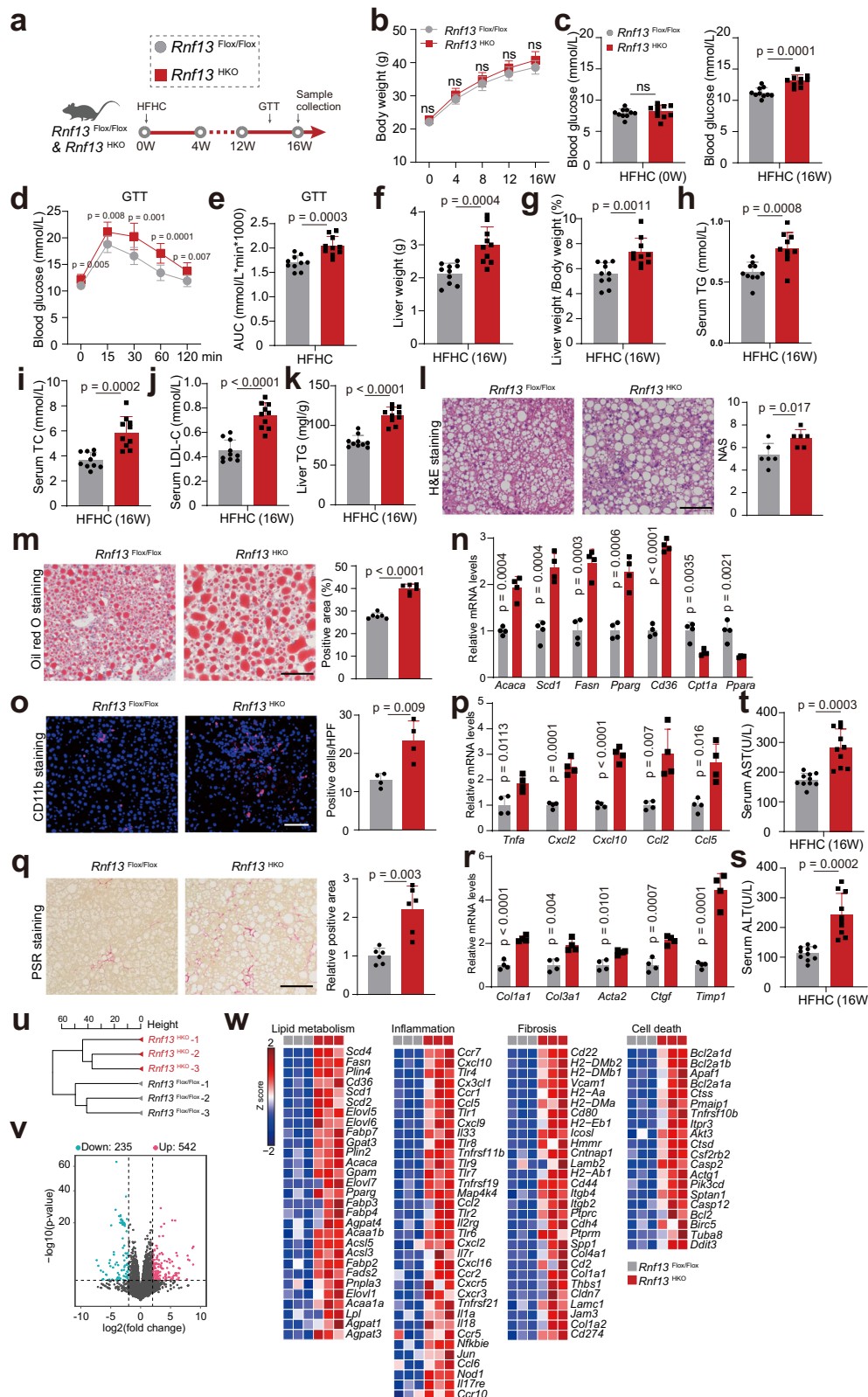

## RNF13 facilitates the degradation of STING in a ubiquitination-dependent way

Subsequently, we investigated how RNF13 suppresses STING protein levels. Given that RNF13 is an E3 ubiquitin ligase, which can modulate the degradation of its substrates, we first performed the CHX chase assay. Results showed that RNF13 shortened the half-life of STING, making it degraded rapidly within 2 h (Fig. 8a). Further, MG132 restored the RNF13-induced decrease of STING abundance, with the CQ treatment conferring marginal change (Fig. 8b). Therefore, it can be inferred that RNF13 facilities the proteasomal degradation of STING. Next, we evaluated whether ubiquitination played a central role in the improvement of NASH by RNF13. We detected enhanced STING ubiquitination in RNF13-overexpressed cells (Fig. 8c). And RNF13 predominantly facilitated K48-linked ubiquitination of STING (Fig. 8d, e).

**Fig. 4 | *Rnf13*ᴴᴷᴼ mice present more severe hepatic steatosis, inflammation and fibrosis in the HFHC model. a** Schematic depiction of in vivo experiments performed to evaluate the function of RNF13 using *Rnf13*ᴴᴷᴼ and *Rnf13*ᶠˡᵒˣ/ᶠˡᵒˣ mice fed with a high-fat and high-cholesterol (HFHC) diet. Body weights (**b**), blood glucose levels (**c**), GTT assay and the corresponding AUC (**d, e**), liver weights (**f**), ratios of liver weight to body weight (**g**), serum TG levels (**h**), serum TC levels (**i**), serum LDL-C levels (**j**) and liver TG levels (**k**) of *Rnf13*ᶠˡᵒˣ/ᶠˡᵒˣ and *Rnf13*ᴴᴷᴼ mice at the indicated time points during HFHC consumption (*n* = 10). H&E (**l**), Oil Red O (**m**), CD11b staining (**o**), PSR staining (**q**) and corresponding quantification of liver sections obtained from *Rnf13*ᶠˡᵒˣ/ᶠˡᵒˣ and *Rnf13*ᴴᴷᴼ mice fed with HFHC for 16 weeks. Scale bars, 100 μm (for **l, m,** and **q**, *n* = 6; for **o**, *n* = 4). Lipometabolic (**n**), proinflammatory (**p**) and profibrotic (**r**) mRNA expression in the liver of *Rnf13*ᶠˡᵒˣ/ᶠˡᵒˣ and *Rnf13*ᴴᴷᴼ mice after HFHC feeding (*n* = 4). Serum ALT (**s**) and AST (**t**) levels of *Rnf13*ᶠˡᵒˣ/ᶠˡᵒˣ and *Rnf13*ᴴᴷᴼ mice after HFHC feeding (*n* = 10). Hierarchical clustering analysis (**u**), volcano plot (**v**) and gene hot map (**w**) showing the results of RNA sequencing using the liver tissues from *Rnf13*ᶠˡᵒˣ/ᶠˡᵒˣ and *Rnf13*ᴴᴷᴼ mice after 16-week HFHC feeding (*n* = 3). Data were expressed as mean ± SD. Two-tailed Student's *t*-test for **c** and **e**–**t**, two-way repeated-measures ANOVA followed by Bonferroni post hoc analyses for **b** and **d**. Source data are provided as a Source Data file.

Consistently, the catalytically inactive mutant of RNF13 failed to affect STING ubiquitination as well as its degradation (Fig. 8f, g), and exerted negligible impacts on NASH phenotypes (Fig. 8h–l). Thus, our data support the role of RNF13 in mediating the ubiquitination-dependent degradation of STING in NASH. However, we did not detect a distinct interaction between RNF13 and STING in co-immunoprecipitation (co-IP) assays (Supplementary Fig. 6a), suggesting that there may exist another E3 ligase, which conducts the RNF13-induced degradation of STING.

## TRIM29 is the downstream effector of RNF13 for STING degradation

Using immunoprecipitation coupled with mass spectrometry (IP-MS) (Fig. 9a), we identified an E3 ligase TRIM29, which has been reported to promote STING degradation in the context of innate immunity[35,36]. Meanwhile, STING did not present in the IP-MS result (Supplementary Fig. 6b). Accordingly, TRIM29 was selected as the candidate target of RNF13. To testify it, we carried out co-IP assays. As expected, there existed a clear interaction between RNF13 and TRIM29 (Fig. 9b). Consistently, the co-localization of RNF13 and TRIM29 was observed via a confocal laser scanning microscope (Supplementary Fig. 6c). Moreover, the in vitro pull-down analysis proved a direct interaction between RNF13 and TRIM29 (Supplementary Fig. 6d). And the molecular mapping assays showed that the transmembrane (TM) domain of RNF13 was responsible for its interaction with the C-terminal domain of TRIM29 (Supplementary Fig. 6e). We then verify whether TRIM29 mediated STING degradation by RNF13. First, TRIM29 interacted (Supplementary Fig. 7a) and then promoted the K48-linked ubiquitination of STING (Fig. 9c and Supplementary Fig. 7b). And RNF13 knockdown by small interfering RNA (si*RNF13*) hampered the K48-linked ubiquitination of STING, which can be restored by TRIM29 overexpression (Fig. 9d). Moreover, adenovirus-mediated TRIM29 overexpression blocked the upregulation of STING (Fig. 9e), the intensified lipid accumulation and inflammation response in Adsh*Rnf13*-infected primary hepatocytes (Supplementary Fig. 7c-g). Taken together, our data prove that RNF13 promotes STING degradation through TRIM29.

## RNF13 stabilizes TRIM29 by enhancing the K63-linked ubiquitination of TRIM29

Thereafter, we investigated how RNF13 regulates TRIM29. We first observed that RNF13 elevated TRIM29 protein level in a dose-dependent manner (Fig. 9f and Supplementary Fig. 8a). Moreover, CHX chase assay showed that RNF13 stabilized TRIM29 (Fig. 9g). In consideration of RNF13 being an E3 ubiquitin ligase, we performed the ubiquitination assay. The result showed that RNF13 considerably promoted TRIM29 ubiquitination (Fig. 9h), whereas TRIM29 marginally impacted on the ubiquitination of RNF13 (Supplementary Fig. 8b). Furthermore, RNF13 specifically facilitated the addition of K63O polyubiquitin chains to TRIM29 (Fig. 9i and Supplementary Fig. 8c). Given that prior studies have demonstrated the role of K63-linked ubiquitination in stabilizing proteins[37,38], it is likely that RNF13 stabilized TRIM29 by enhancing its K63-linked ubiquitination. To testify it, we introduced RNF13 mutant (C258A/H260A) in the following assays. As expected, we observed the inability of RNF13 mutant (C258A/H260A) to catalyze the K63-linked ubiquitination of TRIM29 (Fig. 9j), or controlling the protein abundance of TRIM29 and STING (Fig. 9k). All in all, our data prove that in the context of NASH, RNF13 facilities the K63-linked ubiquitination of TRIM29, which subsequently enhances TRIM29 stability and primes it to accelerate STING degradation through K48-linked ubiquitination (Fig. 9l).

## Discussion

Conventionally speaking, innate immune cells play an essential role during microbial infection. Intriguingly, current studies have proved the enormous impacts of innate immunity system on NAFLD progression: it not only initiates inflammation responses in hepatic tissue but also conducts immune-independent pathologies, such as steatosis and hepatic insulin resistance[5]. Hence, molecules and pathways belonging to innate immune system play an active part in NASH pathogeneses, and an emerging research field called immunometabolism deserves in-depth investigation.

Despite prominent studies investigating the role of TLRs and NLRs in NAFLD[5,39], more attention has been given to the cGAS-STING pathway[40,41], the major cytosolic DNA-sensing pathway of the innate immune system. As indicated in previous studies, human and murine hepatocytes do not express STING, accounting for the weak capacity of hepatocytes to combat HBV infection[42]; the cGAS-STING-TBK1 pathway induced in macrophages, but not hepatocytes, fuels NASH[25,26]. However, as research progresses, novel findings are being successively made. First, several researchers have independently detected the activation of STING signaling pathway in hepatocytes[24,30–32]: upon metabolic stress, the released microbial or mitochondrial DNA can be sensed by cGAS[40]; the STING pathway can also be induced by endoplasmic reticulum (ER) stress[31,32], a key molecular event in NASH[2]. Second, studies have revealed several biological functions of the cGAS-STING-TBK1 pathway in hepatocytes: activated STING signaling pathway contributes to p62 phosphorylation and protein inclusion accumulation in NASH[30]; STING can restrict lipophagy[28] and inhibiting the STING signaling pathway can ameliorate the dyslipidemia and inflammation in FFAs-treated hepatocytes[27,29]. In this study, we also verified the proinflammatory and lipogenic role of STING in hepatocytes by adopting a modified cellular model that more faithfully mimics in vivo NAFLD conditions. Furthermore, we proved STING is the downstream target of RNF13 in NASH, since RNF13 overexpression in hepatocytes can suppress STING protein level as well as the activation of its downstream signaling; the adenovirus-mediated STING overexpression significantly restored severe lipid deposition and inflammatory response in RNF13-overexpressed hepatocytes. Additionally, our study shows that TRIM29, which has been proven to negatively regulate local innate immunity through inducing STING degradation, also participates in NASH pathogenesis.

Indeed, canonical inflammatory cells, such as Kupffer cells, monocytes, and neutrophils exert key regulatory functions in innate immune response. Whereas, hepatocytes and LSECs have also been reported to participate in immunological regulation[10]. The latest study even shows that hepatocytes can be a regulatory hub of recruited macrophages and CD4⁺ T cells upon lipopolysaccharide (LPS)

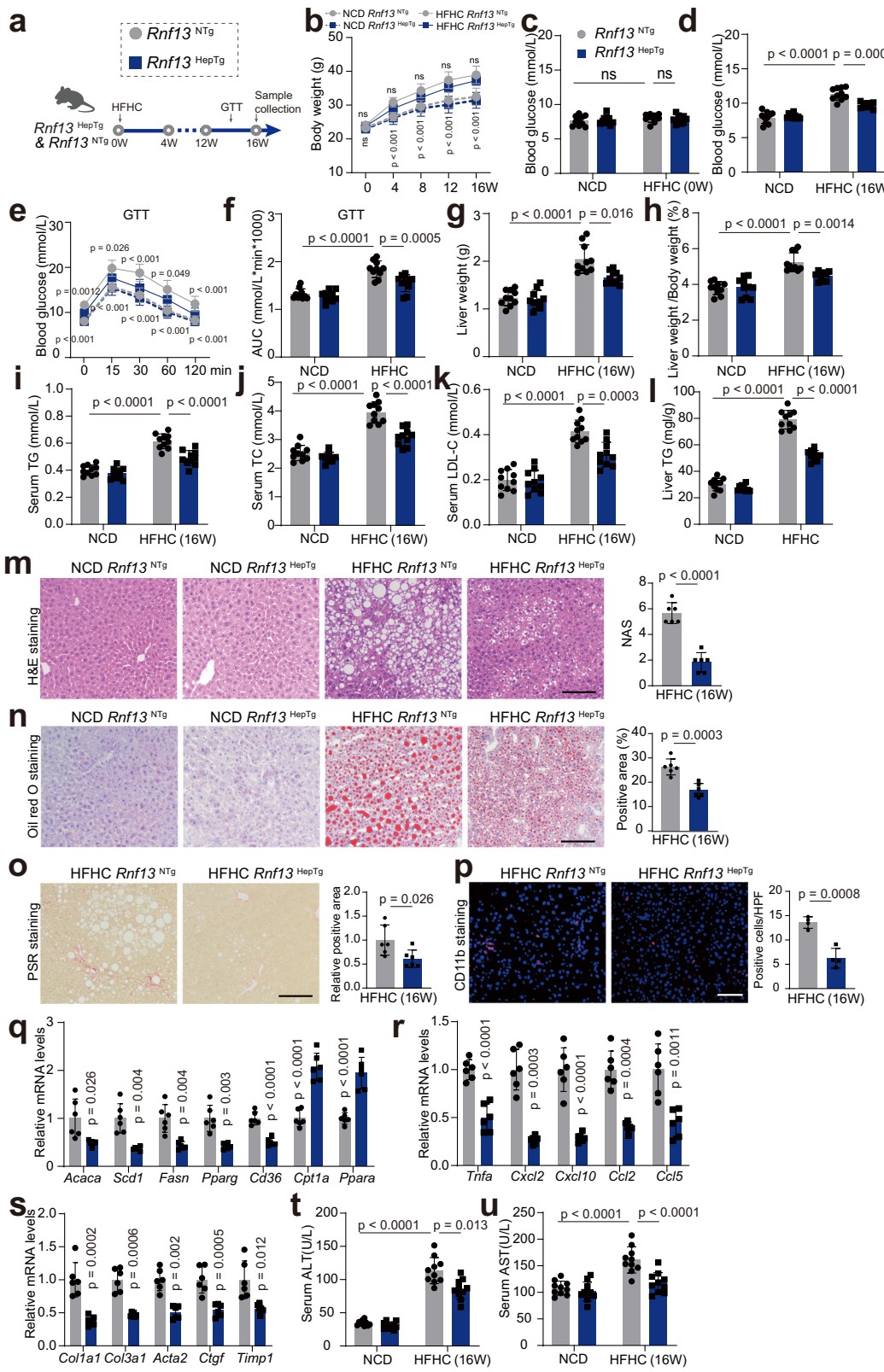

challenge[43]. Therefore, despite the highly expressed STING in inflammatory cells, the endogenous STING pathway in hepatocytes is reasonable to participate in regulating NASH progression. On one hand, the STING pathway regulates the production of cytokines in hepatocytes, as we and others have proved[27,29], which might facilitate the recruitment of nonparenchymal cells and exacerbate lipid deposition as well as the injury of hepatocytes in a vicious circle; on the other hand, given that STING-dependent type I IFN response engages in controlling the synthesis and import of lipids[44], and STING is associated with SCAP-SREBP1 complex[45], it is plausible that STING can directly regulate lipid metabolism in hepatocytes, thus resulting in the initiation of lipotoxicity-induced endoplasmic reticulum (ER) stress, mitochondrial dysfunction, inflammation and so on. Despite some wrinkles that remain to be ironed out, it is reasonable that the STING

**Fig. 5 | Rnf13^HepTg mice present less severe hepatic steatosis, inflammation and fibrosis in the HFHC model. a** Schematic depiction of in vivo experiments performed to evaluate the function of RNF13 using hepatic *Rnf13*-overexpressed transgenic mice (*Rnf13*^HepTg) and non-transgenic mice (*Rnf13*^NTg) fed with a high-fat and high-cholesterol (HFHC) diet. Body weights (**b**), blood glucose levels (**c, d**), GTT assay and the corresponding AUC (**e–f**), liver weights (**g**), ratios of liver weight to body weight (**h**), serum TG levels (**i**), serum TC levels (**j**), serum LDL-C levels (**k**) and liver TG levels (**l**) of *Rnf13*^NTg and *Rnf13*^HepTg mice at the indicated time points during NCD or HFHC consumption (*n* = 10). H&E (**m**), Oil Red O (**n**), PSR staining (**o**), CD11b staining (**p**), and corresponding quantification of liver sections obtained from *Rnf13*^NTg and *Rnf13*^HepTg mice fed with NCD or HFHC for 16 weeks. Scale bars, 100 μm

(for **m–o**, *n* = 6; for **p**, *n* = 4). Lipometabolic (**q**), proinflammatory (**r**), and profibrotic (**s**) mRNA expression in the liver of *Rnf13*^NTg and *Rnf13*^HepTg mice after NCD or HFHC feeding (*n* = 4). Serum ALT (**t**) and AST (**u**) levels of *Rnf13*^NTg and *Rnf13*^HepTg mice after NCD or HFHC feeding (*n* = 10). Data were expressed as mean ± SD. Two-tailed Student's *t*-test for **m–s**, one-way ANOVA with Bonferroni post hoc analysis for **c, d, f–l, t** and **u**, two-way repeated-measures ANOVA followed by Bonferroni post hoc analyses for **b** and **e** (Upper *p*-value for comparison between HFHC *Rnf13*^NTg group and HFHC *Rnf13*^HepTg group; Lower *p*-value for comparison between NCD *Rnf13*^NTg group and HFHC *Rnf13*^NTg group). Source data are provided as a Source Data file.

signaling pathway in hepatocytes is a potent promoter in the transition from NAFL to NASH, and targeting components in this signaling pathway turns out to be a promising therapy for individuals with NASH.

In our work, we identify RNF13 as an effective inhibitor of STING and downstream signaling pathways in NASH pathogenesis. Members of the PA-TM-RING family that RNF13 belongs to have distinct associations with innate immunity, e.g., RNF128, RNF130, RNF150, and RNF204[16–18]. As for RNF13, studies have reported that it induces ER stress and apoptosis through activating the IRE1α-TRAF2-JNK signaling pathway[46,47]. Despite ER stress converting detrimental impacts on the homeostasis of hepatocytes in NAFLD[2], and RNF13 having been reported to be an ER-anchored protein[19], we failed to detect the interaction between RNF13 and vital mediators of ER stress, such as IRE1, PERK, and ATF6, in our IP-MS results. Besides, in exploring the potential signaling pathways and biological processes regulated by RNF13 in NASH, we also failed to detect obvious enrichment of ER stress-related pathways in transcriptome analysis. More importantly, functional assays in our study identify RNF13 as an anti-inflammatory protein in NASH, rather than a proinflammatory one. We speculate that these discrepancies result from the different cell types and treatments, and further studies are needed to fully evaluate the biological functions of RNF13 in different pathological circumstances.

Instead of ER stress-related proteins, we reveal that TRIM29 mediates the regulation of RNF13 on NASH. Prior studies have reported the role of TRIM29 in negatively regulating DNA virus-triggered innate immune response through facilitating STING degradation[35,36]. Notably, we first prove the TRIM29-STING regulatory axis during NASH progression, which also favors that factors involved in innate immunity might be potential therapeutic targets for NASH treatment. Intriguingly, our work presents a dual E3 ligase-dependent regulation in NASH pathogenesis. Previous studies have also documented the combinations among E3 ligases and/or deubiquitinating enzymes (DUBs) in the regulation of cell cycle, autophagy, antiviral response and so on[48,49]. However, it has rarely been reported in NASH. Given the accumulating evidence showing the extensive participation of ubiquitin-based post-translational modifications in NASH, more importance should be attached to the crosstalk among the E3 ligases and DUBs. Our work enriches the theory of dual E3 ligase-mediated regulation, which might contribute to the theoretical foundation of multi-target therapeutics for individuals with NASH.

We admit the limitations of our study. First, considering the impacts of STING on NASH can be rather complicated, as we have discussed above, we predominantly focused on the overall impacts of the RNF13-TRIM29-STING regulatory axis. Further studies are warranted to investigate the way STING fuels NASH. Second, given that *Rnf13* gene disruption in hepatocytes exhibits significant impacts on insulin resistance and blood glucose, whether RNF13 participates in the regulation of insulin signaling pathways in other metabolic organs merits further investigation. Lastly, to further confirm the potential of RNF13 in clinical translation, large animal experiments should be conducted.

In conclusion, we found that RNF13 is a potent inhibitor of lipid deposition, inflammatory response, and metabolic disturbance during NASH progression. Mechanical speaking, RNF13 stabilizes TRIM29 via K63-linked ubiquitination, thus priming TRIM29 to degrade STING in a ubiquitination-dependent way and terminate the abnormal activation of the downstream signaling pathways.

## Methods
### Animal models
Animal experiments were approved by the Animal Care and Use Committee of Fourth Military Medical University. All animals received humane care, according to the criteria outlined in the Guide for the Care and Use of Laboratory Animals prepared by the National Academy of Sciences and published by the National Institutes of Health. C57BL/6 J male mice aged 6-8 weeks were included in this study, and housed in pathogen-free conditions with a 12-hour light/dark cycle and temperature kept at 22−24 °C, humidity kept at 40%-70%. To establish a NAFL model, mice were fed with a high-fat diet (HFD; protein, 20%; fat, 60%; carbohydrates, 20%; H10060; HUAFUKANG Bioscience; Beijing, China) for 24 weeks. To establish a NASH model, mice were fed with a high-fat and high-cholesterol diet (HFHC; protein, 14%; fat, 42%; carbohydrates, 44%; cholesterol, 0.2%; TP 26304; Trophic Diet; Nantong, China) for 16 weeks. The mice in the control group were fed with a normal chow diet (NCD; protein, 18.3%; fat, 10.2%; carbohydrates, 71.5%; 1010001; XIETONG BIO-ENGINEERING; Jiangsu, China) for corresponding time durations. Isoflurane (2%, 0.5 L/min) were used in animal euthanasia practice.

### Cell lines and primary cells
Human embryonic kidney 293 (GNHu43) and 293 T (GNHu17) cells were purchased from the Cell Bank of the Type Culture Collection of the Chinese Academy of Sciences, Shanghai, China. The human liver cancer cell lines Huh7 (GDC0134) and HepG2 (GDC0024) were purchased from the China Center for Type Culture Collection, Wuhan, China. All the cell lines were free of mycoplasma contamination, and were verified before use. Primary hepatocytes and hepatic non-parenchymal cells were isolated from male C57BL/6 male mice aged 8-10 weeks as previously described[50]. After anesthetized, mice were perfused sequentially with the Liver Perfusion Medium (17701-038; Thermo Fisher Scientific; Waltham, MA, USA) and the Liver Digestion Medium (17701-034; Thermo Fisher Scientific; Waltham, MA, USA) through the inferior vena cava. The liver was isolated, minced, and then filtered through a cell strainer (70 μm). Hepatocytes were obtained by centrifuging the solution at 50 *g* for 4 min at 4 °C. The supernatants were then centrifuged at 350 *g* for 10 min and resuspended in DMEM containing 10% FBS. Then, the 25% and 50% Percoll (17-0891-02; GE-Healthcare) separation followed by centrifugation at 350 *g* for 10 min at 4 °C were conducted, and Kupffer cells as well as hepatic sinusoidal endothelial cells were isolated from the stratified liquid. Phosphate-buffered saline (PBS) was used to wash the cells for three times before they were seeded. After incubation for 20 min, the supernatant containing endothelial cells was collected and seeded in a new plate, while Kupffer cells remained on the former plate. All of

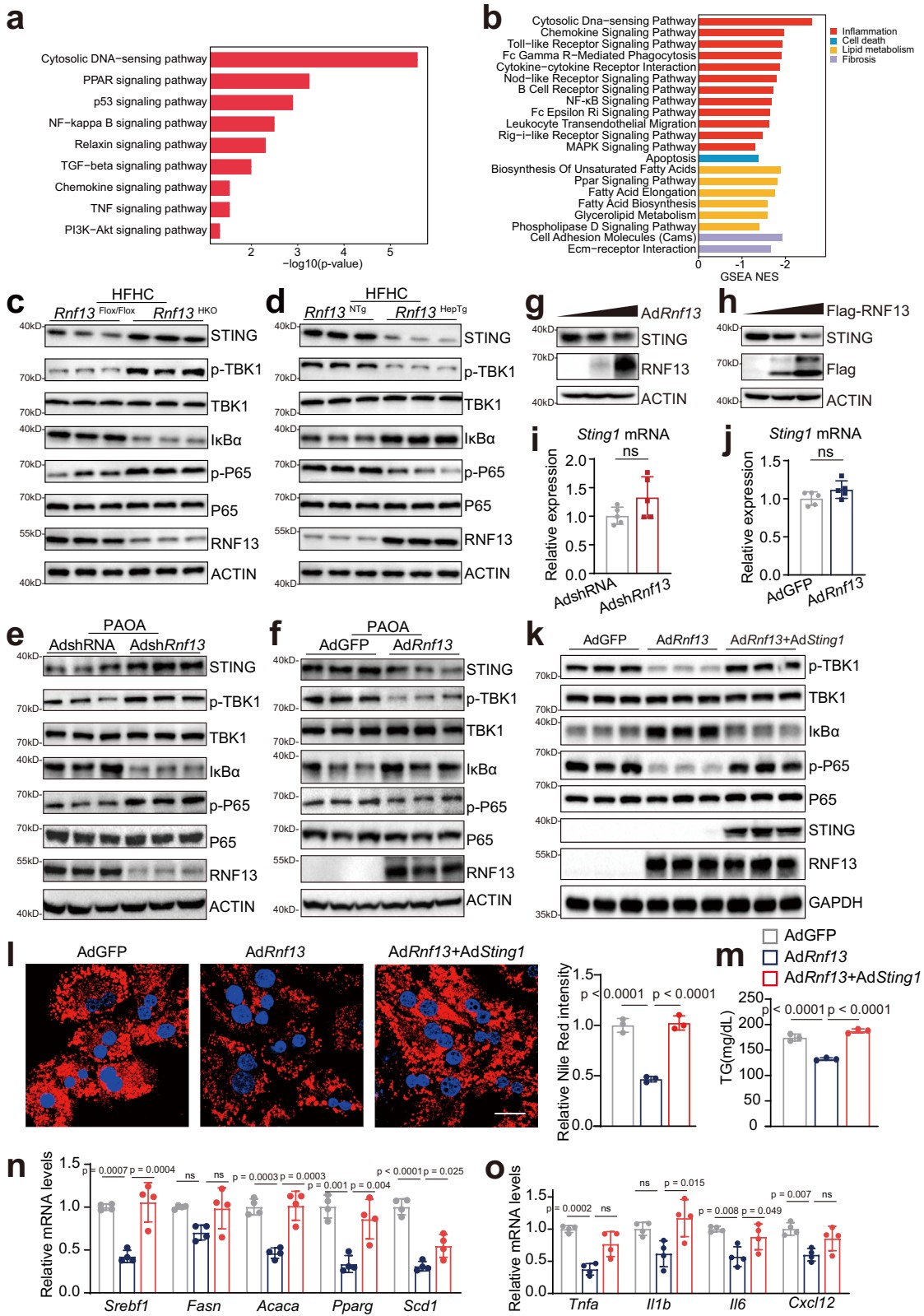

**Fig. 6 | RNF13 inhibits STING-mediated inflammatory signaling pathways in NASH.** KEGG (**a**) analysis and GSEA (**b**) showing the results of RNA sequencing using the liver tissues from *Rnf13*^Flox/Flox and *Rnf13*^HKO mice after 16-week HFHC feeding. The Indicated protein levels in the livers of HFHC-induced *Rnf13*^Flox/Flox and *Rnf13*^HKO mice (**c**), and *Rnf13*^NTg and *Rnf13*^HepTg mice (**d**) (*n* = 3). The Indicated protein levels in MPHs infected with Adsh*Rnf13* (**e**) and Ad*Rnf13* (**f**) as well as the corresponding control viruses followed by PAOA treatment for 12 h (*n* = 3). Endogenous STING protein level in MPHs (**g**) and HepG2 cells (**h**) in response to different doses of RNF13 overexpression. qPCR analyses of *Sting1* mRNA in MPHs infected with Adsh*Rnf13* (**i**) and Ad*Rnf13* (**j**) as well as their corresponding control viruses, followed by PAOA treatment for 12 h (*n* = 5). The indicated protein levels (**k**), Nile Red staining and quantification (**l**), TG contents (**m**), lipogenic (**n**) and proinflammatory gene expression (**o**) in MPHs infected with AdGFP, Ad*Rnf13* or Ad*Rnf13* plus Ad*Sting1* with PAOA treatment for 12 h (For **k**–**m**, *n* = 3; for **n**–**o**, *n* = 4). Scale bars, 25 μm. Data were expressed as mean ± SD. Two-tailed Student's *t*-test for **i** and **j**, one-way ANOVA with Bonferroni post hoc analysis for **l**–**o**. Source data are provided as a Source Data file.

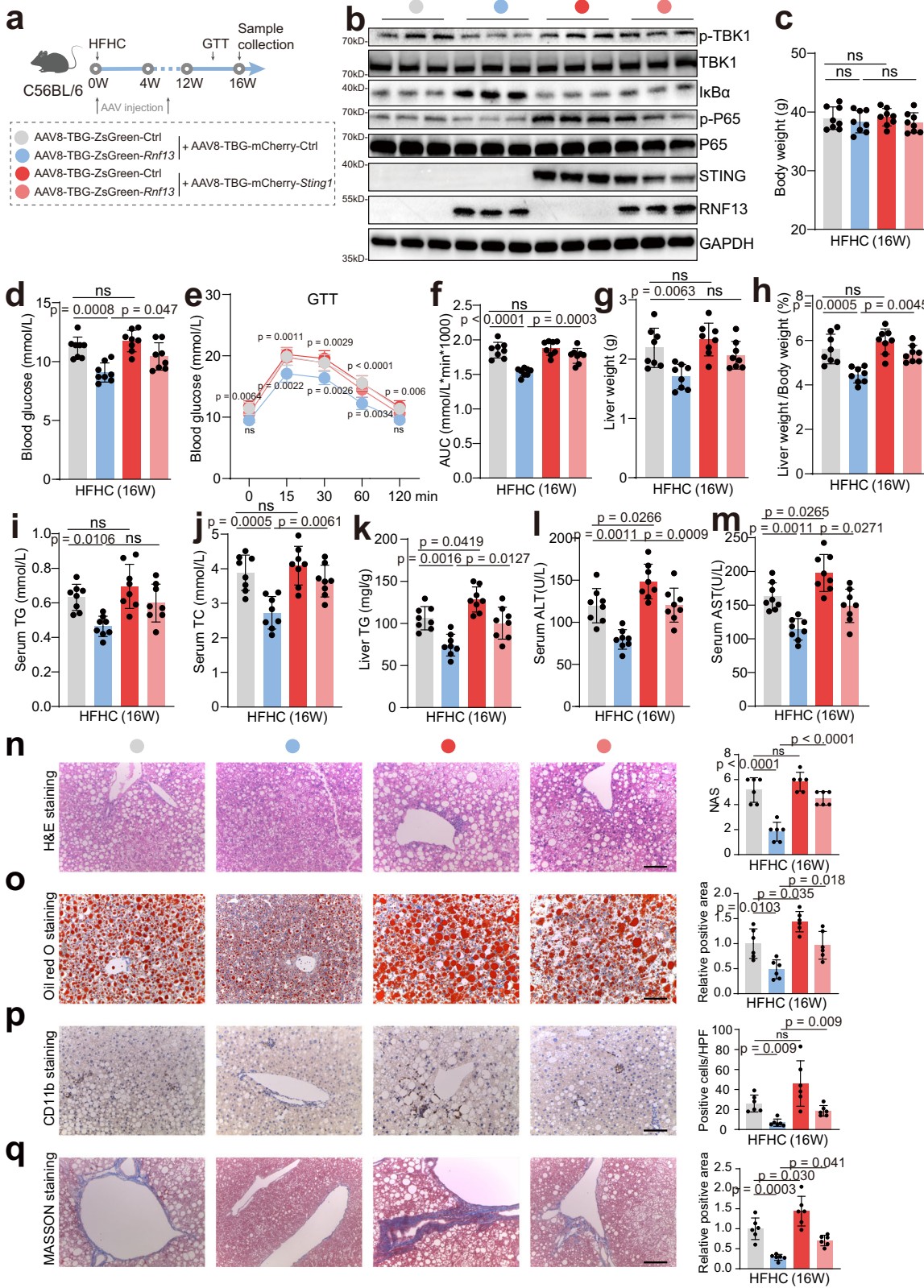

**Fig. 7 | In vivo experiments confirm the regulatory effects of RNF13 on STING in NASH. a** Schematic depiction of the in vivo rescue experiments. **b** The protein level of RNF13, STING, p-P65/P65, IκBα and p-TBK1/TBK1 in the livers of the mice from the indicated groups (*n* = 3). Body weights (**c**), blood glucose levels (**d**), GTT assays (**e**) and the corresponding AUC (**f**), liver weights (**g**), ratios of liver weight to body weight (**h**), serum TG (**i**), serum TC (**j**), liver TG (**k**), serum ALT (**l**) and AST levels (**m**) of the mice from the indicated groups (*n* = 8). H&E (**n**), Oil Red O (**o**), CD11b (**p**) and

Masson (**q**) staining and corresponding quantification of liver sections of the mice from the indicated groups (*n* = 6). Scale bars, 100 μm. Data were expressed as mean ± SD. One-way ANOVA with Bonferroni post hoc analysis for **c, d,** and **f–q**, two-way repeated-measures ANOVA followed by Bonferroni post hoc analyses for **e** (Upper *p*-value for comparison between *Rnf13*-overexpressed/OE group and the control group; Lower *p*-value for comparison between *Rnf13*-OE group and *Rnf13*&*Sting1*-OE group). Source data are provided as a Source Data file.

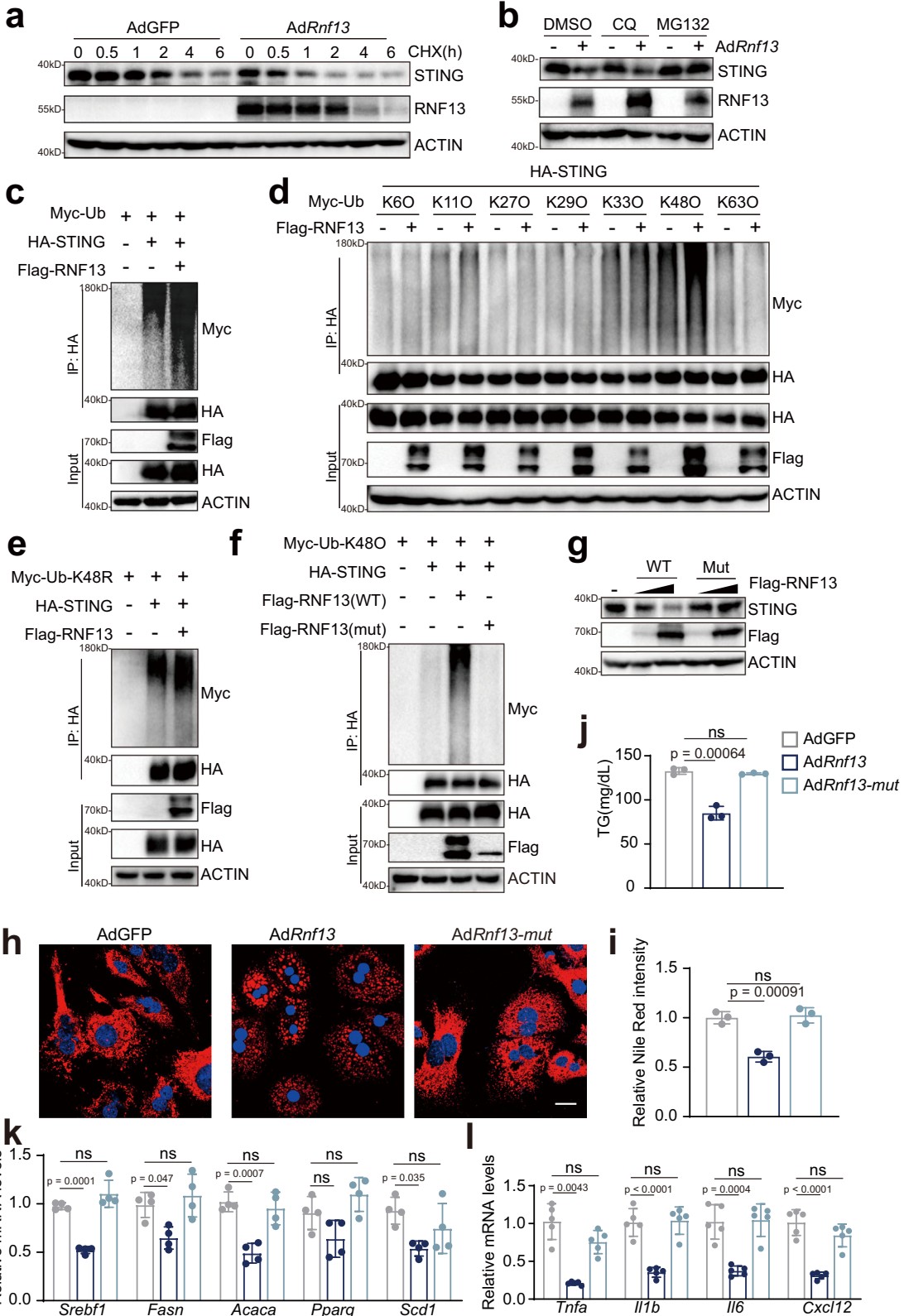

**Fig. 8 | RNF13 facilitates the degradation of STING in a ubiquitination-dependent way. a** STING protein expression in MPHs infected with AdGFP or Ad*Rnf13*, followed by PAOA plus CHX treatment. **b** STING protein expression in MPHs infected with AdGFP or Ad*Rnf13*, followed by PAOA plus CQ or MG132 treatment. **c-f** Ubiquitination of HA-STING in HepG2 cells co-transfected with the indicated plasmids, followed by PAOA plus MG132 treatment. **g** STING protein expression in HepG2 cells transfected with different doses of Flag-RNF13 (wild type) or Flag-RNF13 (mutant) expressing plasmids, followed by PAOA treatment. Nile Red staining (**h**) and quantification (**i**), TG contents (**j**), lipogenic (**k**) and proinflammatory gene expression (**l**) in MPHs infected with AdGFP, Ad*Rnf13* (wild type) or Ad*Rnf13* (mutant), followed by PAOA treatment (For **h**–**j**, *n* = 3; for **k**, *n* = 4; for **l**, *n* = 5 samples). Data were expressed as mean ± SD. One-way ANOVA with Bonferroni post hoc analysis for **i-l**. For **a-l**, at least three independent experiments have been conducted. Source data are provided as a Source Data file.

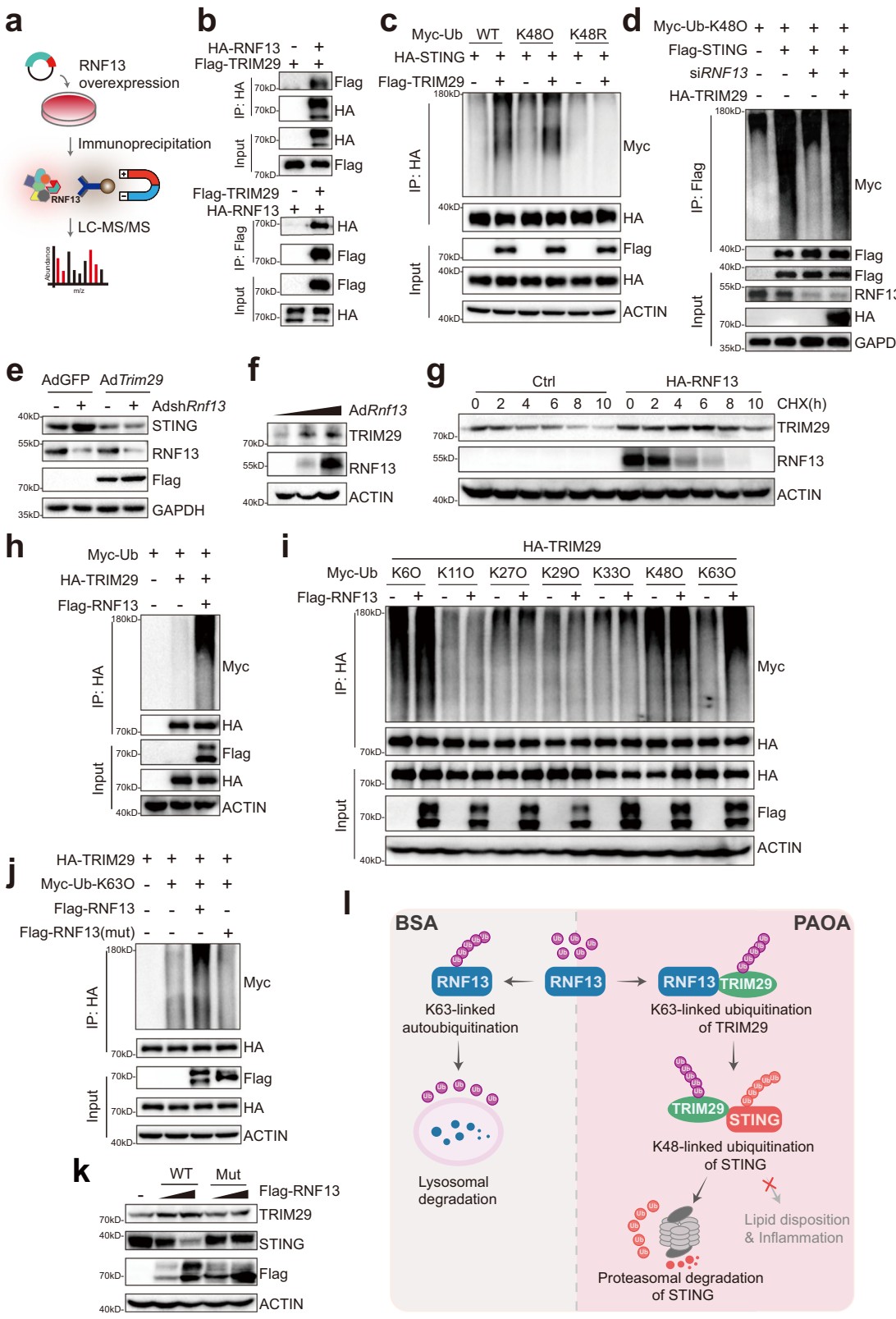

the cells were cultured in Dulbecco's modified Eagle's medium (DMEM; 10569010; Gibco; Carlsbad, CA, USA) containing 10% fetal bovine serum (F05-001-B160216; Bio-One Biotechnology; Guangzhou, China) and 1% penicillin-streptomycin (15140-122; Gibco by Invitrogen; Carlsbad, CA, USA) in a cell incubator with 5% $CO_2$ at 37 °C. To inhibit the proteasomal or lysosomal degradation, cells were treated with 25 mM MG132 (S2619; Selleck; Houston, TX, USA) or

25 mM chloroquine (CQ; S6999; Selleck; Houston, TX, USA) for 12 h respectively. 100 μg/ml cycloheximide (CHX; S7418; Selleck; Houston, TX, USA) was used in the CHX chase assay. 1 μM C176 (S6575; Selleck; Houston, TX, USA) was added to the medium for 18 h to inhibit the activation of STING. Kupffer cells were treated with 1 ng/ml TNF-α (AF-315-01A-20, PeproTech; Cranbury, NJ, USA) to examine RNF13 expression.

**Fig. 9 | TRIM29 is the downstream effector of RNF13 for STING degradation.**
**a** Schematic depiction of the workflow of IP-MS. **b** Co-immunoprecipitation of Flag-TRIM29 and HA-RNF13 in HEK293T cells co-transfected with the indicated plasmids. **c, d** Ubiquitination of exogenous STING in HepG2 cells co-transfected with the indicated plasmids and si*RNF13*, followed by PAOA plus MG132 treatment. **e** STING protein expression in MPHs infected with Ad*Trim29* or Adsh*Rnf13* and their corresponding controls, followed by PAOA treatment. **f** TRIM29 protein expression MPHs in response to different doses of RNF13 overexpression. **g** TRIM29 protein expression in HepG2 cells transfected with HA-RNF13 or control plasmids, followed by PAOA plus CHX treatment. **h–j** Ubiquitination of HA-TRIM29 in HepG2 cells co-transfected with the indicated plasmids, followed by PAOA plus MG132 treatment. **k** TRIM29 and STING protein expression in HepG2 cells transfected with different doses of Flag-RNF13 (wild type) or Flag-RNF13 (mutant) expressing plasmids, followed by PAOA treatment. **l** Schematic showing the mechanism of RNF13 degradation and its protective role in response to PAOA treatment. For **b–k**, at least three independent experiments have been conducted. Source data are provided as a Source Data file.

## Human liver samples

Human liver samples were acquired from individuals who underwent hepatic surgery at the Department of Hepatobiliary Surgery, Xijing Hospital of the Fourth Military Medical University, and Zhongnan Hospital of Wuhan University. All the patients enrolled in the present study were excluded from drug or toxin injury, hepatitis virus infection, and excessive alcohol consumption. To evaluate the NAFLD activity score (NAS), at least two pathologists independently went through the hematoxylin and eosin (H&E) staining of liver sections in a blinded fashion based on the NASH-CRN scoring system[51]. Cases that possessed a NAS of more than 4 were included in the NASH group. The collection and application of human samples were approved and supervised by the ethics committee of Xijing Hospital of Fourth Military Medical University and Zhongnan Hospital of Wuhan University, and adhered to the principles listed in the Declaration of Helsinki. All patients have signed an informed consent form in the present study. The basic clinical characteristics and histological assessments are listed in Supplementary Table 1.

## In vitro lipotoxic model

To establish an in vitro lipotoxic model, hepatocytes were treated with 0.2 mM palmitic acid (PA; P0500; Sigma-Aldrich; St. Louis, MO, USA) and 1.0 mM oleic acid (OA; O-1008; Sigma-Aldrich; St. Louis, MO, USA), dissolved in 0.5% fatty acid-free bovine serum albumin (BSA; BAH66-0100; Equitech Bio; Kerrville, TX, USA), for 6–24 h. In a modified cellular model, mouse primary hepatocytes as well as nonparenchymal cells (mostly Kupffer cells) were isolated from the same mouse. Primary hepatocytes were seeded in 6-well plates, while NPCs were seeded on transwell chambers in 24-well plates. After 6 h, hepatocytes were infected with adenovirus for about 12 h. Thereafter, the culture medium containing adenovirus was removed and hepatocytes were washed three times with PBS. Then culture medium containing PAOA and transwell chambers loading NPCs were added to the 6-well plates containing hepatocytes. After another 8–12 h, transwell chambers were removed and hepatocytes were collected for further analyses.

## Generation of genetically modified mice

*Rnf13*^Flox/Flox^ mice were generated using a CRISPR/Cas9 system in the C57BL/6 background. An online CRISPR design tool (http://chopchop.cbu.uib.no/) was used to design two single guide RNAs (sgRNA1 and sgRNA2, listed in Supplementary Table 2) targeting *Rnf13* introns 4 and 5. The loxp-flanked *Rnf13* exon 5 was introduced into the left and right homology arms (1122 bp & 1005 bp) to generate the donor vector for the homology-mediated end joining. The mRNA encoding the Cas9 nuclease, two sgRNAs, and the donor vector were injected into zygotes from C57BL/6 mice. Zygotes were further transplanted into a surrogate female mouse. To confirm that the Flox allele was functional, we conducted in vitro Cre-loxP-mediated recombination with genomic DNA. The primers P2 and P6 were used to detect the deletion products, while the primers P3 and P4 were used to detect the circle products. In order to generate the homozygous *Rnf13*^Flox/Flox^ mice, the identified founder mice were mated with C57BL/6 J mice, with the primers P4 and P5 used to screen progeny. Then, *Rnf13*^Flox/Flox^ mice were crossed to albumin-Cre transgenic mice (003574; Jackson Laboratory; Bar Harbor, ME, USA) to generate hepatocyte-specific *Rnf13*-knockout

(*Rnf13*^HKO^) mice. All products were confirmed by sequencing and primers for mouse genotyping are listed in Supplementary Table 2. For the generation of *Rnf13*^HepTg^ mice, a Sleeping Beauty (SB) transposase system was applied as previously described[52]. In brief, a liver-specific pT3 plasmid carrying *Rnf13* (pT3-EF1a-3xflag-h-*RNF13*) (50 µg per mouse) and the SB100X transposase plasmid (3 µg per mouse) were injected into mouse livers via the tail vein. To overexpress RNF13 and STING in hepatocytes, we adopted two kinds of adeno-associated virus, namely AAV8-TBG-ZsGreen-*Rnf13* and the AAV8-TBG-mCherry-*Sting1*, which were constructed by Hanbio Tech (Shanghai, China). A thyroxine-binding globulin (TBG) promoter was used to achieve hepatocyte-specific overexpression[53]. The virus ($1 \times 10^{11}$ genomic copies/mouse) was injected via tail vein into mice before and after 8 weeks of HFHC feeding.

## Glucose tolerance test (GTT) and insulin tolerance test (ITT)

GTT tests were conducted after 22 weeks of HFD feeding and after 14 weeks of HFHC feeding. ITT tests were conducted after 23 weeks of HFD feeding. After fasting for 6 h, the GTT or ITT assays were carried out. The mice were injected with 1 g/kg glucose or 0.75 IU/kg insulin intraperitoneally, and the level of blood glucose concentrations was measured at the indicated time points. Subsequently, the areas under the curve (AUC) were calculated by the use of the conventional trapezoid rule.

## Biochemical analysis

Commercial kits were used to measure the contents of triglyceride (TG) in primary hepatocytes (290-63701; Wako; Tokyo, Japan), according to the manufacturer's instructions. The serum concentrations of alanine aminotransferase (ALT), aspartate aminotransferase (AST), TG, TC, and LDL-C were measured by an ADVIA 2400 Chemistry System Analyzer (Siemens, Tarrytown, NY, USA), according to the manufacturer's instructions.

## Histological and immunohistochemical staining

Liver tissues were fixed with 10% formaldehyde for 48 h. H&E staining (Hematoxylin, G1004; Servicebio; Wuhan, China; Eosin, BA-4024; Baso; Zhuhai, China) was conducted on paraffin-embedded liver sections to visualize histological features. Oil Red O staining (O0625; Sigma-Aldrich; St. Louis, MO, USA) was conducted on OCT-embedded liver sections to evaluate hepatic steatosis. Picrosirius red staining (PSR; 26357-02; Hede Biotechnology; Beijing, China) was conducted to assess liver fibrosis. To detect RNF13 expression in liver tissues, the paraffin-embedded liver slides were incubated with the anti-RNF13 primary antibody at 4 °C overnight. Then, slides were incubated with the Rabbit Two-step Detection Kit (Rabbit Enhanced Polymer Detection System, PV-9001; ZSGB-BIO; Beijing, China). The positive cells were visualized after DAB staining (ZLI-9018; Zhongshan Biotech; Beijing, China). The histological and immunohistochemical images were acquired with a light microscope (ECLIPSE 80i; Nikon; Tokyo, Japan).

## Immunofluorescence staining

To evaluate the infiltration of macrophages in the liver tissues, paraffin-embedded liver slides were first labeled with the primary antibody, anti-CD11b, at 4 °C overnight. Then the slides were incubated

with a fluorophore-conjugated secondary antibody. The immuno-fluorescence images were obtained by using a fluorescence micro-scope (Olympus; Tokyo, Japan). At least seven high-power fields (HPF) of each animal were randomly selected to qualify the infiltration of inflammatory cells. For coverslip staining, cells transfected with the indicated plasmids were fixed with 4% paraformaldehyde for 10 min at room temperature, and then cells were permeabilized with 0.2% Triton for 10 min at room temperature. After blocking with 10% BSA for 1 h at room temperature, cells were incubated with anti-Flag, anti-HA tag, or anti-LAMP1 antibody at 4 °C overnight. Then cells were labeled with fluorophore-conjugated secondary antibodies. DAPI (S36939; Invitro-gen; Carlsbad, CA, USA) was used to stain the nuclei. Images were acquired with a confocal laser scanning microscope (TCS SP8; Leica; Wetzler, Germany), according to the manufacturer's instructions. All antibodies used in this study are listed in Supplementary Table 5.

### Nile Red staining
Adenovirus-infected murine primary hepatocytes were stimulated with PAOA for 8-12 h. After being washed three times by PBS, the cells were fixed with 4% paraformaldehyde for 10 min at room temperature. Then cells were stained with Nile Red (1 µM in PBS; 22190; Fanbo Biochemicals; Beijing, China) for 10 min at room temperature. Nuclei were visualized with DAPI. Images of lipid disposition were visualized and quantified by a laser scanning confocal microscope (TCS SP8; Leica; Wetzler, Germany) or a high-content analysis system (Perki-nElmer; Waltham, MA, USA).

### Plasmid construction and viral infection
For in vitro assays, full-length, truncated, or mutant cDNA sequences of RNF13, TRIM29, and STING were inserted into the pcDNA5 or phage vector by using PCR-based cloning. For the construction of adenoviral vectors, the shuttle plasmid pENTR-U6-CMV-flag-T2A-EGFP and Vira-Power Adenoviral Expression System (V493-20; Invitrogen; Carlsbad, CA, USA) were used. After linearized by PacI (R0547L; NEB; MA, USA), the recombinant adenoviral vector was transfection into 293 cells with polyethyleneimine (PEI; 24765-1; Polysciences; Warrington, UK) trans-fection reagent. After 6-7 days, cells were harvested to obtain the initial adenovirus. Then the adenovirus was amplified by infecting 293 cells with the crude viral lysate, and purified by cesium chloride (CsCl) density gradient centrifugation. The titer was measured by the 50% tissue culture infective dose (TCID50) method. The adenovirus infec-ted mouse primary hepatocytes at a multiplicity of infection (MOI) of 50. The primer sequences used for plasmid construction are listed in Supplementary Table 3.

### mRNA isolation and qPCR assay
The total mRNA of cells and tissues was extracted with TRIzol reagent (T9424; Sigma-Aldrich; St. Louis, MO, USA), and was quantified by Nanodrop 2000 spectrophotometer (Thermo Fisher). Then the RNA was reverse-transcribed into cDNA by the use of the HiScript II Q RT SuperMix for qPCR (containing a gDNA wiper) (R223-01, Vazyme; Nanjing, China), according to the manufacturer's instructions. The abundance of mRNA was measured by ChamQ SYBR qPCR Master Mix (Q311-02, Vazyme; Nanjing, China), in a Real-Time PCR System (Light-Cycler 480 Instrument II, Roche; Basel, BS, Switzerland), following the manufacturer's instructions. The mRNA abundance of the target genes was normalized to *ACTB* (human) or *Actb* (mouse). The primers used in this study are listed in Supplementary Table 4.

### RNA sequencing and data processing
For RNA sequencing, total RNA was extracted from liver or cell samples using TRIzol reagent (T9424; Sigma-Aldrich; St. Louis, MO, USA) as indicated above. Then 200 ng of RNA input per sample and a MGIEasy RNA Library Prep Kit (1000006383; MGI Tech; Shenzhen, China) were used to construct cDNA libraries, according to the manufacturer's

instructions. Single-end libraries were sequenced using MGISEQ 2000 (MGI Tech; Shenzhen, China). For data processing, HISAT2 software (version 2.1.0)[54] was used to map the sequences from clean reads to Ensembl mouse (mm10/GRCm38) reference genomes. SAMtools software (version 1.4)[55] was used to sort and convert the mapped reads to the Binary Alignment Map (BAM) files. StringTie software (version 1.3.3b)[56] was used to calculate the fragments per kilobase of exon model per million mapped fragments (FPKM) values of each gene. DESeq2 software (version 1.2.10)[57] was used to analyze differential gene expression. Genes with a fold change greater than 1.5 and corre-sponding adjusted *p* values less than 0.05 were identified as DEGs.

### Hierarchical clustering analysis
In hierarchical clustering analysis, the R function "hclust" based on an unweighted average distance algorithm was used to construct a phy-logenetic tree of samples, and the z score method was used to nor-malize gene expression levels of each biological replicate.

### Gene set enrichment analysis (GSEA)
For GSEA, each KEGG pathway and involved genes were defined as a gene set. GSEA was conducted on the Java GSEA (version 3.0) platform[58], and the 'Signal2Noise' metric was adopted to generate a ranked list and a 'gene set' permutation type. Gene sets with an FDR value of less than 0.25 were considered statistically significant.

### KEGG pathway enrichment analysis
KEGG pathway enrichment analysis was performed using Fisher's exact test with our in-house R script. The KEGG pathway annotations were downloaded from the KEGG database. Pathways with a *p*-value < 0.05 were considered as significantly enriched pathways.

### Small-interfering RNA transfection
Cells were transfected with siRNA (GenePharma; Shanghai, China) specifically targeting *RNF13* by using Lipofectamine™ 3000 (L3000008; Invitrogen; Carlsbad, CA, USA), according to the manu-facturer's instructions. After transfected for 48 h, the cells were har-vested and the subsequent assays were performed. The antisense sequence of siRNA targeting on *RNF13* is as follows (5'−3'): AUUAACACGAUGAAAGUGCTT.

### Western blot analysis
To perform western blot analysis, tissues and cells were lysed with RIPA lysis buffer (P0013E; Beyotime Biotechnology; Shanghai, China) containing protease inhibitor cocktail tablets (04693132001; Roche; Basel, BS, Switzerland) and phosphatase inhibitor tablets (4906837001; Roche; Basel, BS, Switzerland). A BCA Protein Assay Kit (23225; Thermo Fisher Scientific; Waltham, MA, USA) was used to quantify the concentration of total protein. Then, equal quantities of the indicated protein were separated by 8–12% SDS-PAGE gels and then transferred to PVDF membranes (IPVH00010; Millipore; Bill-erica, MA, USA). After being blocked with 5% skim milk in Tris-buffered saline/Tween 20 (TBST) for 1 h at room temperature, the PVDF membranes with proteins were incubated with the indicated primary antibodies at 4 °C overnight and incubated with HRP-conjugated secondary antibodies for 1 h at room temperature. Sig-nals were then visualized using an ECL kit (170-5061; Bio-Rad; Her-cules, CA, USA) in a ChemiDoc MP Imaging System (Bio-Rad; Hercules, CA, USA). GAPDH or ACTIN was used as a loading control. The antibodies used are listed in Supplementary Table 5.

### Immunoprecipitation and mass spectrometry analysis
Cells transfected with the indicated plasmids for 24 h were lysed with IP lysis buffer (20 mM Tris-HCl, pH 7.4; 150 mM NaCl; 1 mM EDTA; and 1% NP-40) containing protease inhibitor cocktail tablets (04693132001; Roche; Basel, BS, Switzerland) and phosphatase

inhibitor tablets (4906837001; Roche; Basel, BS, Switzerland) for 30 min at 4 °C. The samples were centrifuged at 12,000 $g$ for 10 min, and then incubated with protein A/G agarose beads (11719394001, 11719386001; Roche; Basel, BS, Switzerland) and the indicated primary antibodies at 4 °C overnight. The beads were then washed about three times with the buffer containing 150 mM or 300 mM NaCl. The immunocomplexes were eluted in SDS loading buffer and subjected to the immunoblotting analysis mentioned above. For mass spectrometry (MS) analysis, the eluate was separated by 8-12% SDS-PAGE gels and stained with a Pierce Silver Stain Kit (24612; Thermo Fisher Scientific; Waltham, MA, USA) according to the manufacturer's instructions. The bands were excised and the proteins were subjected to liquid chromatography–tandem mass spectrometry (LC-MS/MS) analysis was carried out by Applied Protein Technology (Shanghai, China). Candidate molecules were selected based on the following criteria: 1) the candidates should be presented in the group of anti-HA immunoprecipitation but absent in the group of anti-IgG immunoprecipitation; 2) the number of unique peptides should be > 2.

### Ubiquitination assays

Cells were transfected with the indicated plasmids for 24 h. Each sample was lysed in 100 μl 10% SDS lysis buffer and then heated at 95 °C for 10 min to be denatured. 0.9 ml cold IP lysis buffer (20 mM Tris-HCl, pH 7.4; 150 mM NaCl; 1 mM EDTA; and 1% NP-40) were added to the lysates. After ultrasonic processing, samples were subjected to centrifugation (12,000 $g$ for 10 min). Then the supernatants were extracted and incubated with indicated primary antibodies as well as protein A/G agarose beads (11719394001, 11719386001; Roche; Basel, BS, Switzerland) at 4 °C for 3 h. The beads were then washed with IP lysis buffer three times. After centrifugated, the beads were boiled with SDS loading buffer for 10 min and the proteins were eluted. Finally, western blotting was performed as mentioned before.

### Statistics and reproducibility

The data in most figure panels reflect experiments performed using independent samples. Statistical analyses were conducted using SPSS software (version 23.0). All data are presented as the mean ± SD values. For comparisons between two groups, the two-tailed Student's $t$-test (for data showing a normal distribution) or the nonparametric Mann-Whitney U test (for data showing a skewed distribution) was performed. For comparisons among multiple groups, one-way ANOVA was performed, followed by the Bonferroni post hoc test (for data showing homogeneity of variance) or Tamhane's T2 post hoc test (for heteroscedastic data). To determine the statistical differences of repeated measurement data, two-way repeated-measures ANOVA followed by the Bonferroni post hoc test was conducted. A $p$-value of less than 0.05 was considered statistically significant. All western blot and micrographs of cellular experiments were repeated at least three times from independent samples with similar results.

### Reporting summary

Further information on research design is available in the Nature Portfolio Reporting Summary linked to this article.

## Data availability

All data are available in the main text or the supplementary materials. Source data are provided within this paper. The RNA-Seq data generated in this study have been deposited in the National Center for Biotechnology Information BioProject database under accession code PRJNA1020209. Source data are provided with this paper.

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

## Acknowledgements

We appreciate Professor Hua Han from Fourth Military Medical University for his technical assistance. This work was supported by grants from the National Key Research and Development Program of China (2021YFA1100500 to L.W., 2016YFA0102100 to L.W.) and the National Natural Science Foundation of China (82325007 to L.W., 81770560 to L.W., 82300684 to Y.H.).

## Author contributions

L.W., Z.L., and P.Y. conceptualized the work. L.W., Z.L., P.Y., and Y.H. designed the experiments. Z.L., P.Y., Y.H., H.X., J.D., and F.H. performed the experiments and analyzed the data. Z.L. and P.Y. wrote the manuscript. L.W. and K.D. reviewed and edited the manuscript. L.W. supervised the project.

## Competing interests

The authors declare no competing interests.
