## [Peer Review File · Nature Communications]

RING finger protein 13 protects against nonalcoholic steatohepatitis by targeting STING-relayed signaling pathwaysEditorial Note: Parts of this Peer Review File have been redacted as indicated to remove third-party material where no permission to publish could be obtained.

REVIEWER COMMENTS

Reviewer #1 (Remarks to the Author):

NAFLD is one of the most common forms of liver disease in Western industrialized nations and is strongly associated with both insulin resistance and metabolic syndrome. Finding protective molecular is important as this may provide new approaches for therapeutic intervention. In this study, the authors demonstrate a protective role for RNF13 for NAFLD/NASH. They propose that RNF13 leading to insulin resistance and hepatic steatosis by inhibiting STING pathway. These are interesting observations. The major concerns on this work include:

- 1.To confirm the impacts of RNF13 on STING pathway in NASH, what about the expression of key proteins of STING pathway in the livers of mice with HFD and HFHC diet. And knockout mouse should be used in this manuscript.
 - 2.STING is a stimulator of interferon genes. The interferon levels should be measured in Vivo and in Vitro.
 - 3.Please add information about biological replicates used in Fig.2, Fig.6, Fig.7.
 - 4.Methods are poorly described. For eg: the gender of mice is missing. Where were these mice from?
 - 5.Moreover, it remains unclear, whether the experiments were carried out in a blinded fashion.
 - 6.Oil-red O staining should be accompanied with quantitative measurement of hepatic triglycerides by an enzymatic assay.
- Hallmarks of NASH include hepatocyte ballooning and fibrosis. Please have a pathologist evaluate for hepatocyte ballooning.

Reviewer #2 (Remarks to the Author):

In this study, Lin et al. showed that hepatic RNF13 protein was upregulated in the liver of

individuals with NASH. Moreover, RNF13 protein level was positively correlated with NASH severity measured by the NAFLD activity score (NAS). The authors found that Rnf13 depletion in hepatocyte exacerbated insulin resistance, steatosis, inflammation, cell injury and fibrosis in liver, which could be relieved by Rnf13 overexpressing. RNF13 alleviated NASH progression by inhibiting the cGAS-STING pathway. This is an interesting manuscript which investigated the protective role of RNF13 in NASH.

Here are some major concerns:

1. The authors confirmed that RNF13 expression level was increased in NASH, but overexpression of RNF13 was shown to improve NASH in experiments. How to explain this phenomenon? What causes the increased expression of RNF13 in NASH? In addition, the authors suggested that RNF13 protein level positively correlated with NASH severity measured by the NAFLD activity score (NAS). I don't think that's an accurate statement.
2. The authors confirmed that the degradation of RNF13 was inhibited under PA/OA treatment. What is the underlying mechanism?
3. In animal experiments, the regulatory effect of RNF13 on STING should be confirmed by rescue experiments after overexpression of RNF13.
4. The authors confirm that there is an interaction between RNF13 and TRIM29. Is there an interaction between TRIM29 and STING? The interaction between TRIM29 and STING is not shown in Figure 8C.
5. In Fig 8, the author studied the ubiquitination effect of RNF13 on TRIM29 by introduced RNF13 mutant (C258A/H260A). In addition to TRIM29 protein expression, STING expression should also be detected.
6. Why RNF13 facilitates the K63-linked 295 ubiquitination of TRIM29 to enhance TRIM29 stability? However, RNF13 facilitated K48-linked ubiquitination of STING to promote STING degradation? How to explain this phenomenon?
7. As for modified cellular model, the details were not clear.

Minor concerns:

1. In Fig. 1B, there are multiple #1 in the NASH group, please check if they are correct.
2. How does RNF13 change from NAFLD to NASH? Has it increased?
3. In Fig.4 and Fig.5, there was no data related to insulin resistance, such as ITT, etc.

Response to Reviewer 1

General comment

NAFLD is one of the most common forms of liver disease in Western industrialized nations and is strongly associated with both insulin resistance and metabolic syndrome. Finding protective molecular is important as this may provide new approaches for therapeutic intervention. In this study, the authors demonstrate a protective role for RNF13 for NAFLD/NASH. They propose that RNF13 leading to insulin resistance and hepatic steatosis by inhibiting STING pathway. These are interesting observations.

Response:

We are grateful for the positive comments from the reviewer.

Major Comment 1

To confirm the impacts of RNF13 on STING pathway in NASH, what about the expression of key proteins of STING pathway in the livers of mice with HFD and HFHC diet. And knockout mouse should be used in this manuscript.

Response:

We thank the reviewer for this thoughtful suggestion. To further confirm the impacts of RNF13 on STING pathway during NAFLD progression, we detected the expression of key proteins of STING pathway in the livers of *Rnf13*^{HKO}, *Rnf13*^{HepTg} and the control mice fed with HFD or HFHC diet. Consistent with the results from the in vitro experiments, we observed that *Rnf13* knockout in hepatocytes led to an increase in the protein level of STING, p-TBK1 and p-p65, and a decrease in I κ B α level. While RNF13 overexpression in liver hampered the activation of STING signaling. **The results are presented below and in Supplementary Fig. 4d and Fig. 6c, d.**

As the reviewer suggested, the *Sting1* knockout mice is important to explore the role of STING in NAFLD. Whereas, we did not generate the knockout mice, since several excellent studies have well demonstrated the vital role of hepatocyte STING in NAFLD pathogenesis: ① Donne et al. found that replication stress in NAFLD hepatocyte is sufficient to elicit DNA lesions and to drive the activation of the cGAS-STING pathway (*Developmental Cell*, 2022, PMID: 35768000); ② Liu et al. demonstrated that hepatocyte STING activation increase lipid accumulation by regulating MTOR activity in the progression of hepatic steatosis (*Autophagy*, 2022, PMID: 34382907); ③ Cho et al. revealed that the hepatocyte cGAS-STING axis mediates saturated fatty acids-induced activation of TBK1, which provokes the accumulation of insoluble inclusion bodies and NASH progression (*Hepatology*, 2018, PMID: 29251796). Besides, despite it's a good suggestion to generate the *Sting1* hepatocyte-specific knockout mice, it would take almost one year to fulfill it. Hence, we would consider to use the mice in our future works.

Instead of *Sting1* knockout mice, we generated the *Sting1* hepatocyte-specific overexpressing (STING-OE) mice via adeno-associated virus serotype 8 (AAV8). The AAV8 carries plasmids containing a thyroxine-binding globulin (TBG) promoter, which guarantees the hepatocyte-specific *Sting1* overexpression (*Cell Stem Cell*, 2022, PMID: 36055192). After HFHC feeding for 16 weeks, we observed that STING-OE mice showed higher levels of ALT, AST and hepatic TG, and more severe liver steatosis, inflammation and fibrosis compared with the control mice. However, STING overexpressing has no impacts on body weight,

liver weight, blood glucose, serum TG and TC levels. **The results are presented in Fig. 7, Supplementary Fig. 4j-l and below.**

In conclusion, we and others have revealed the deleterious effects of hepatocyte STING in NASH. Moreover, we proved RNF13 suppresses the STING pathway in the NASH liver.

Major Comment 2

STING is a stimulator of interferon genes. The interferon levels should be measured in Vivo and in Vitro.

Response:

Agree. Since STING directly activates the transcription of type I IFNs, we measured the mRNA level of interferon- β (IFN- β) in vivo and in vitro. Results showed that RNF13 knockout significantly facilitated the transcription of IFN- β . Likewise, RNF13 overexpressing reduced the mRNA level of IFN- β . **The results are presented below and in Supplementary Fig. 4e-i.**

Major Comment 3

Please add information about biological replicates used in Fig.2, Fig.6, Fig.7.

Response:

Thanks for the suggestion. We have added information about biological replicates in these figures.

Major Comment 4

Methods are poorly described. For eg: the gender of mice is missing. Where were these mice from?

Response:

In fact, we delineated the methodology of this study in the Supplementary Materials. And now we have moved it to the Revised Manuscript. Please check it.

Major Comment 5

Moreover, it remains unclear, whether the experiments were carried out in a blinded fashion.

Response:

All of the animal experiments and a part of in vitro experiments were carried out in a blinded fashion. And all of the in vitro experiments were independently performed by 2-3 individuals, for at least 3 times.

Major Comment 6

Oil-red O staining should be accompanied with quantitative measurement of hepatic triglycerides by an enzymatic assay. Hallmarks of NASH include hepatocyte ballooning and fibrosis. Please have a pathologist evaluate for hepatocyte ballooning.

Response:

As the reviewer suggested, hepatic triglycerides were measured. **The results are presented below and in Fig. 3m, 4k and 5l.**

Actually, we have evaluated the progression of NAFLD by measuring the NAFLD Activity Score (NAS) on H&E-stained liver sections. The NAS contains three items, namely steatosis, lobular inflammation and hepatocyte ballooning (*Hepatology*, 2005, PMID: 15915461), as the Table 1 describes (*The EPMA Journal*, 2014, PMID: 25937854). And below the Table 1 is one example of our assessment of NAS (Fig. 5m).

[REDACTED]

Group	Number	steatosis	inflammation	ballooning	NAS
Control-HFHC	C3412	3	1	1	5
	C3416	3	1	1	5
	C3420	3	1	2	6
	C3421	3	2	2	7
	C3424	3	1	2	6
	C3425	3	1	1	5
RNF13-HFHC	C3403	1	0	1	2
	C3404	2	0	0	2
	C3405	2	0	1	3
	C3409	1	0	0	1
	C3410	2	0	0	2
	C3413	1	0	0	1

Response to Reviewer 2

General comment:

In this study, Lin et al. showed that hepatic RNF13 protein was upregulated in the liver of individuals with NASH. Moreover, RNF13 protein level was positively correlated with NASH severity measured by the NAFLD activity score (NAS).

The authors found that Rnf13 depletion in hepatocyte exacerbated insulin resistance, steatosis, inflammation, cell injury and fibrosis in liver, which could be relieved by Rnf13 overexpressing. RNF13 alleviated NASH progression by inhibiting the cGAS-STING pathway. This is an interesting manuscript which investigated the protective role of RNF13 in NASH.

Response:

We appreciate the reviewer's positive comments for our work.

Major Comment 1

The authors confirmed that RNF13 expression level was increased in NASH, but overexpression of RNF13 was shown to improve NASH in experiments. How to explain this phenomenon? What causes the increased expression of RNF13 in NASH? In addition, the authors suggested that RNF13 protein level positively correlated with NASH severity measured by the NAFLD activity score (NAS). I don't think that's an accurate statement.

Response:

In our opinion, the level of a cellular-protective molecular does not always decrease in pathological circumstance. Our work has proved that RNF13 is upregulated during NASH progression. Such upregulation might be a reaction to various stimulations of NASH. However, under persist pathological stimulations, other NASH-promoting signaling pathways could also be activated, and upregulated RNF13 is insufficient to ameliorate NASH. Therefore, we could overexpress RNF13 via Sleeping Beauty Transposase or adeno-associated virus to achieve its protective function. Similar situations can be seen in our work and other's: TRIM16, which is upregulated in response to lipotoxicity, ameliorates lipid accumulation and inflammation during NASH progression (*Cell Metabolism*, 2021, PMID: 34146477); TRIM16, which is upregulated during cardiac hypertrophy, alleviates phenylephrine-induced cardiac hypertrophy (*Circulation Research*, 2022, PMID: 35437018).

The reason accounting for increased expression of RNF13 is that saturated fatty acids inhibit its lysosomal degradation. Previous study shows that RNF13 protein undergoes extensive post-translational proteolysis in both lysosome and proteasome, which makes it rather unstable (**Figures A-B from *The FEBS Journal*, 2009, PMID: 19292867**). ① In our work, we did not observed an increase of RNF13 mRNA level in NASH (Fig. 1d, f, h and j), hence we considered its increase results from the enhanced stability of RNF13 protein. And this hypothesis was proved by the CHX (which can inhibit protein synthesis) chasing assays, in which we observed that remaining RNF13 protein started to degrade after synthesis inhibition, and RNF13 protein degraded much more slowly upon saturated fatty acids stimulation (Fig. 2a). ② Generally speaking, the protein degradation is governed by lysosome or proteasome. Therefore, we adopted lysosome inhibitor (chloroquine, CQ) and proteasome inhibitor (MG132) to investigate the way RNF13 degrades. Results showed that CQ can rescue RNF13 protein level in the presence of CHX (Fig. 2b, c), indicating that RNF13 mainly undergoes lysosomal rather than proteasomal degradation in the setting of saturated fatty acids stimulation. ③ Further, we proved that CQ, rather than MG132, can increase RNF13 level in BSA-treated hepatocytes to the degree of saturated fatty acids stimulated (Fig. 2e), indicating that the inhibited lysosomal degradation is the reason accounting for the increased RNF13 level in NASH. ④ We then investigated the reason for inhibited lysosomal degradation in NASH. Previous studies indicate that RNF13 undergoes intense auto-ubiquitination (*Cell Research*, 2009, PMID: 18794910), and ubiquitination has been documented to directs internalized proteins toward lysosome (*Cell*, 2010, PMID: 21111229), we therefore wondered whether the lysosomal degradation of RNF13 was governed by ubiquitination. Subsequently, we found K63-linked auto-ubiquitination of RNF13 was inhibited upon saturated fatty acids stimulation (Fig. 2g), and deprive RNF13 of its ubiquitination could make it rather stable even without saturated fatty acids treatment (Fig. 2i). Thus, the inhibited K63-linked auto-ubiquitination by PAOA accounts for the attenuated lysosomal degradation of RNF13. All in all,

RNF13 undergoes extensive K63-linked auto-ubiquitination and subsequent lysosomal degradation in normal circumstances; whereas saturated fatty acids stimulation can alleviate the auto-ubiquitination and lysosomal degradation of RNF13, making it upregulated in NASH.

Indeed, NAS could not accurately reflect NASH severity. Hence, we would just state “RNF13 protein level is positively correlated with NAFLD activity score (NAS)”.

[REDACTED]

(Figures A-B from *The FEBS Journal*, 2009, PMID: 19292867)

Major Comment 2

The authors confirmed that the degradation of RNF13 was inhibited under PA/OA treatment. What is the underlying mechanism?

Response:

Our work revealed that under PA/OA treatment, the attenuated K63-linked auto-ubiquitination of RNF13 inhibits its lysosomal localization and degradation. Numerous of E3 ligase, such as RNF13, could induce ubiquitination on themselves, which could affect their stability, trafficking or activation. In our work, we observed that under normal circumstance (BSA treatment), RNF13 could undergo intense K63-linked auto-ubiquitination (Fig. 2g), which was inhibited by PA/OA treatment. Previous studies have revealed that K63-linked ubiquitination could induce lysosomal degradation of the protein (*Cell*, 2010, PMID: 21111229; *Nature Communications*, 2021, PMID: 34285233). Thus, we conducted the rescue experiments to prove the causal relationship between RNF13 K63-linked auto-ubiquitination and lysosomal degradation. In doing this, we use a ubiquitination-nulled mutant RNF13 C258A/H260A, in which cysteine (C258) and histidine (H260) in the ring-finger domain were mutated to alanine. This mutant could not induce K63-linked auto-ubiquitination (Fig. 2h) and has a longer half-life than wild type RNF13 in BSA treatment (Fig. 2i). Thus, we have demonstrated that PA/OA treatment inhibits K63-linked auto-ubiquitination of RNF13, which

consequently blocks its lysosomal degradation and then increases its protein abundance.

Major Comment 3

In animal experiments, the regulatory effect of RNF13 on STING should be confirmed by rescue experiments after overexpression of RNF13.

Response:

Thanks for this constructive suggestion. Actually, we have been preparing the in vivo rescue experiments after the first submission. Methodologically, we constructed the AAV8-TBG-ZsGreen-*Rnf13* and the AAV8-TBG-mCherry-*Sting1* (AAV8, adeno-associated virus serotype 8; TBG, a thyroxine-binding globulin promoter, which guarantees the hepatocyte-specific overexpression), to generate control, RNF13-, STING- and RNF13&STING-overexpressed mice. After 16-week HFHC feeding, we observed that RNF13 overexpressing significantly attenuated the abnormal blood glucose, lipid accumulation in serum as well as liver, inflammation response and fibrosis. And the therapeutic effects were abolished in the RNF13-STING overexpressing mice, indicating that RNF13 ameliorates NASH through regulating STING. **The results are presented in Fig. 7, Supplementary Fig. 4j-l and below.**

Major Comment 4

The authors confirm that there is an interaction between RNF13 and TRIM29. Is there an interaction between TRIM29 and STING? The interaction between TRIM29 and STING is not shown in Figure 8C.

Response:

In fact, the interaction between TRIM29 and STING has been presented in Supplementary Fig. 5a (in the first submission). And we have stated it clearly in the Revised manuscript, “First, TRIM29 interacted (Supplementary Fig. 6a) and then promoted the K48-linked ubiquitination of STING (Fig. 9c)”.

Major Comment 5

In Fig 8, the author studied the ubiquitination effect of RNF13 on TRIM29 by introduced RNF13 mutant (C258A/H260A). In addition to TRIM29 protein expression, STING expression should also be detected.

Response:

Actually, we have already measured the impacts of RNF13 wild type and mutant on STING expression (Fig. 8g). In spite of this, we would take the reviewer’s advice and detect STING expression again. **The results are presented below and in Fig. 9k.**

Major Comment 6

Why RNF13 facilitates the K63-linked 295 ubiquitination of TRIM29 to enhance TRIM29 stability? However, RNF13 facilitated K48-linked ubiquitination of STING to promote STING degradation? How to explain this phenomenon?

Response:

At first, we found RNF13 facilitated K48-linked ubiquitination of STING, but to our surprise, we did not detect a direct interaction between RNF13 and STING (Generally, E3 ligase, like RNF13, would directly interact with its substrates). Hence, we performed IP-MS and find a “bridge” between RNF13 and STING, namely TRIM29. In the subsequent experiments, we found that RNF13 can facilitate the K63-linked ubiquitination of TRIM29 to enhance its stability and protein abundance. And TRIM29 can facilitate K48-linked ubiquitination of STING to promote its degradation. Thus, RNF13 first enhances K63-linked ubiquitination of TRIM29 to enhance TRIM29 stability; more TRIM29 can then facilitate K48-linked ubiquitination of STING to promote STING degradation; in general, RNF13 facilitated K48-linked ubiquitination of STING.

Major Comment 7

As for modified cellular model, the details were not clear.

Response:

We appreciate this kind reminding. In the modified cellular model, mouse primary hepatocytes as well as nonparenchymal cells (mostly Kupffer cells) were isolated from the same mouse. Primary hepatocytes were seeded in 6-well plates, while NPCs were seeded on transwell chambers in 24-well plates. After 6 hours,

hepatocytes were infected with adenovirus for about 12 hours. Thereafter, the culture medium containing adenovirus was removed and hepatocytes were washed for three times with PBS. Then culture medium containing PA/OA and transwell chambers loading NPCs were added to the 6-well plates containing hepatocytes. After another 12 hours, transwell chambers were removed and hepatocytes were collected for further analyses. The flow chart is presented below. We would elaborate it in the Methods section.

Minor Comment 1

In Fig. 1B, there are multiple #1 in the NASH group, please check if they are correct.

Response:

The #1 sample was used as the control sample to quantify RNF13 protein level among 32 human liver samples. We would state it in the figure legend.

Minor Comment 2

How does RNF13 change from NAFLD to NASH? Has it increased?

Response:

Thanks for this thoughtful suggestion. We compared RNF13 protein level in the liver of the NCD-, HFD- and HFHC-induced mice, and detected the highest level

in the HFHC group, indicating that RNF13 protein does increase from NAFL to NASH. The results are presented below and in Fig. 1e.

Minor Comment 3

In Fig.4 and Fig.5, there was no data related to insulin resistance, such as ITT, etc.

Response:

In fact, we have performed ITT before in HFHC-induced RNF13 knockout, RNF13 overexpressing mice and their counterparts (Fig.4 and Fig.5), **and the results are presented below**. Nevertheless, as has been pointed out in the review (*Hepatology*, 2019, PMID: 30372785), HFHC diet is not suitable for studying insulin resistance. Therefore, we only showed the results of GTT (which focuses on the sensitivity of insulin to glucose), instead of ITT (which focuses on the sensitivity of tissues to insulin), in Fig.4 and Fig.5.

REVIEWERS' COMMENTS

Reviewer #1 (Remarks to the Author):

This article proposes that STING is a target of RNF13, so it is crucial to verify the role of RNF13 in NFALD by inhibiting the function of STING. The function of STING can be inhibited using STING inhibitors or methods such as siRNA. Both STING inhibitors and siRNA can be purchased in the market.

Reviewer #2 (Remarks to the Author):

The author has answered my questions very well and suggested publication.

Response to Reviewer 1

Comment

This article proposes that STING is a target of RNF13, so it is crucial to verify the role of RNF13 in NAFLD by inhibiting the function of STING. The function of STING can be inhibited using STING inhibitors or methods such as siRNA. Both STING inhibitors and siRNA can be purchased in the market.

Response:

We deeply appreciate this suggestion. To further confirm the RNF13-STING axis in hepatocytes during NAFLD progression, we applied C176, a highly selective inhibitor of STING (*Nature*, 2018, PMID: 29973723), in the rescue assays. Results showed that the phenotypic changes resulted from RNF13-knockdown can be reversed by C176. Results are presented below and in **Supplementary Fig. 5a-d**.

Response to Reviewer 2

Comment:

The author has answered my questions very well and suggested publication.

Response:

We gratefully appreciate the reviewer's constructive comments that substantially improve our work!